

# ABJ anomaly as a U(1) symmetry and Noether's theorem

**Valentin Benedetti⋆, Horacio Casini† and Javier M. Magan‡**

Instituto Balseiro, Centro Atómico Bariloche, 8400-S.C. de Bariloche, Río Negro, Argentina

⋆ valentin.benedetti@ib.edu.ar , † horaciocasini@gmail.com , ‡ javier.magan@cab.cnea.gov.a

## Abstract

The Adler-Bell-Jackiw anomaly determines the violation of chiral symmetry when massless fermions are coupled to an abelian gauge field. In its seminal paper, Adler noticed that a modified chiral $U(1)$ symmetry could still be defined, at the expense of being generated by a non-gauge-invariant conserved current. We show this internal $U(1)$ symmetry has the special feature that it transforms the Haag duality violating sectors (or non local operator classes). This provides a simple unifying perspective on the origin of anomaly quantization, anomaly matching, applicability of Goldstone theorem, and the absence of a Noether current. We comment on recent literature where this symmetry is considered to be either absent or non-invertible. We end by recalling the DHR reconstruction theorem, which states 0-form symmetries cannot be non-invertible for $d > 2$, and argue for a higher form-symmetry reconstruction theorem.



# 1 Introduction

In this work we revisit an old topic in QFT, namely the Adler-Bell-Jackiw (ABJ) anomaly [1,2], together with its interplay with an even older topic, namely Noether's theorem [3]. The ABJ or chiral anomaly originally refers to the anomalous decay of the neutral pion, when compared with current algebra predictions [4,5]. This same anomaly appears in 4d QED and in many other generalizations. A massless fermion coupled to electromagnetism contains a naively conserved chiral current $J_5^\mu = \bar{\psi}\gamma^5\gamma^\mu\psi$. If conserved, this current would generate a $U(1)$ global symmetry. But the chiral current is a composite operator, and one needs to define it properly through regularization. This process famously breaks the conservation of the chiral current and leads to

$$\partial_\mu J_5^\mu = \frac{1}{16\pi^2}\,\epsilon^{\mu\nu\alpha\beta}F_{\mu\nu}F_{\alpha\beta}\,, \tag{1}$$

where $F$ is the field strength of the electromagnetic field, with a normalization such that $\psi$ has unit charge. The anomaly can be proven in several different ways, see [6].

On the other hand, Noether's theorem asserts the correspondence between conserved charges and symmetries that leave the action invariant. In the context of QFT this theorem typically gets enhanced to a correspondence between continuous symmetries and conserved local currents $j^\mu$. Below we call this enhancement the strong version of Noether's theorem, as opposed to the weak version, where only local implementations of the symmetry (twist operators), acting on compact regions, are required. The weak version has been proved under very general conditions [7–10]. A long standing question in QFT has been to determine under what conditions the strong version holds. Recently, we have made a Lagrangian independent proposal to answer this question [11]. Using the algebraic approach [12,13] to generalized symmetries [14], it simply asserts that the strong version of Noether's theorem is violated for a continuous global symmetry whenever there are generalized symmetries charged under it. To be more precise, the theory has to contain Haag duality violating (HDV) operator classes/sectors that are not invariant under the symmetry. HDV classes appear when there are operators associated with a certain region that cannot be locally generated in the region. Equivalently the theory contains "non-locally generated operators" and in an abuse of language we will often simply call them "non local operators". It is not difficult to prove that the existence of a Noether current implies the HDV classes are invariant under such symmetry [11]. The conjecture is the more subtle opposite implication. An important byproduct of such investigation is that it explains the violation of Noether's theorem in the Weinberg-Witten theorem [15], rederiving it from the perspective of generalized symmetries.

Now, already in its seminal paper [1], Adler mentioned that a seemingly harmless, but insightful modification of the chiral current would give rise to an understanding of the anomaly in terms of a conventional $U(1)$ global symmetry. Namely defining a new local current $\tilde{J}_5$ by

$$\tilde{J}_5{}^\mu = J_5{}^\mu - \frac{1}{8\pi^2}\epsilon^{\mu\nu\alpha\beta}A_\nu F_{\alpha\beta} \quad \rightarrow \quad \partial_\mu \tilde{J}_5{}^\mu = 0\,, \tag{2}$$

one gets a conventional conservation equation. The problem, acknowledged by Adler, is that $\tilde{J}_5$ is not a gauge-invariant operator and the theory does not have a conserved current operator associated with such symmetry. Still, by integrating the local charge density over a Cauchy slice in Minkowski space, one arrives at a gauge invariant conserved charge, generating a $U(1)$ global symmetry group. In this perspective, the ABJ anomaly just redefines the chiral symmetry.

However, this context raises several important questions that form the motivations of this paper. The first is that if we take this modified $U(1)$ chiral symmetry seriously, then this class of theories seems to violate the strong version of Noether's theorem.[1] Then, if the proposal developed in [11] holds, these theories must display HDV sectors charged under the new chiral symmetry. We will show this is indeed the case. Our conclusion is that the modified $U(1)$ symmetry is an ordinary internal symmetry of the theory with the peculiar feature that it mixes the HDV classes of the electromagnetic field. This peculiarity explains the absence of the Noether current. We remark that the way in which the symmetry acts on the non local sectors was not imagined in our previous paper [11]. There we assumed a general non trivial action of a continuous symmetry on non local classes would lead to non compact non local sectors, and presumably to a free theory. Hence, the anomaly gives an interesting example of non trivial action on the non local classes, that, in principle, seem compatible with an interacting theory. We will further comment on this below and develop a classification of the type of possible actions.

Quite insightfully, this picture provides a new perspective on the origin of the quantization of the anomaly. Briefly, such quantization is forced upon the compatibility of the $U(1)$ cycles associated with the correct chiral group and the group of non local operators. Equivalently, we find that the anomaly on the symmetry and the fact that a symmetry changes the HDV classes is the same physical phenomenon. This perspective also explains anomaly matching [16], including the existence of massless excitations in the infrared model, in a conventional (symmetry based) manner. Lastly, this perspective allows the applicability of Goldstone's theorem.

This subject has attracted much recent attention, and somewhat similar ideas and computations have been advanced and studied in several recent papers [17–19] (see also [20–28] for further related developments). While we agree with the relevant computations on these papers, our conclusions differ appreciably. In particular, we find that neither the chiral symmetry is reduced to $Z_{n_f}$ in QED with $n_f$ massless Dirac fields [17], nor it can be interpreted as non invertible [18, 19]. The origin of the discrepancies is to be attributed to particular misconceptions concerning the local physical manifestations of the generalized symmetries.[2] The essence of the phenomenon are some "textures" on the distribution of local operator algebras in spacetime (of any topology). These textures appear in any compact subregion of any spacetime and can be formalized as Haag duality violations. The unfortunate terminology used in the area precisely hides this very same phenomenon. Therefore, a second motivation for this work is an attempt to clarify the overall physical picture in comparison with recent literature.

---

[1]We thank D. Harlow and H. Ooguri for the suggestion to study anomalous models that challenge Noether's theorem.

[2]These misconceptions are quite uniform in the generalized symmetry literature. They are sometimes harmless, explaining why they have been overlooked. But their consequences get amplified in these subtle scenarios in which different symmetries mix with each other, as happens in anomalous models.

A third motivation is the following. While the existence of a Noether current forbids a non trivial action of the symmetry on the non local classes, the contrary implication requires the construction of a current when all possible non local classes are invariant under the symmetry. This necessarily involves the physics of the UV, equivalently a consistent QFT completion of the theory. In [11] we argued (without mathematical rigor) that such a construction suggested itself from the existence of arbitrarily small twists, together with particular properties these twists possess under the previous assumptions. The anomalous models give an interesting playground for checking these ideas. In particular, an anomalous current seems difficult to repair in a gauge invariant way, but the particular action of the chiral symmetry on non local classes can be challenged in a number of ways. One can explore dimensions different from $d = 4$ where the topology does not allow for such non trivial actions. One can also attempt to destroy the non local classes by introducing other charges [17]. In the cases we analyzed we find that either the chiral symmetry does not exist anymore or the theory seems quite hopeless of a UV completion.

The next three sections follow the line of the three motivations described above. In the discussion section we end by recalling the DHR reconstruction theorem [29–33], highlighting the difficulties, or rather the impossibility, it poses to have genuine internal global (0-form) non invertible symmetries, for dimensions $d > 2$. Our results concerning theories with ABJ anomalies are therefore consistent with such general theorem. We further outline the idea of a "generalized reconstruction theorem" according to which all possible non invertible one-form generalized symmetries, for dimensions $d > 3$,[3] have to be the result of an abelian symmetry quotiented by a global symmetry group.

# 2 ABJ anomaly as a $U(1)$ symmetry

In this section, we study scenarios with abelian electromagnetic anomaly. We start by describing the case of pion electrodynamics since in this effective model the anomaly is manifested at the classical level. All the features of the symmetry, including its action on local and non local operators are very explicit and simple to picture. Then we describe the case of massless QED, where we essentially adapt the discussion by Adler [1], supplemented by the Witten effect [34] that determines the action of the symmetry on the 't Hooft loops.[4] Both pion and QED models tell the same story, that of a continuous group of internal symmetries that transform non trivially the HDV classes. We will highlight how the quantization of the anomaly arises here from the compatibility between the two $U(1)$ cycles associated with the two symmetries, equivalently from the compatibility between the 0-form modified chiral symmetry and the 1-form magnetic symmetry. 't Hooft's anomaly matching between the UV and IR physics is derived in this light as the result of the existence of an ordinary $U(1)$ global symmetry. We then recall how the non trivial action on HDV classes prevents the existence of a conserved current. Finally, we start developing an abstract classification of the possible types of non trivial actions of a $U(1)$ symmetry on non local operators.

## 2.1 Pion electrodynamics

In the context of the effective theory of the pion field, a more transparent Lagrangian approach can be used, where the anomaly follows from the equations of motion. Let us then discuss first

---

[3]For $d = 3$, the topology of sectors corresponding to 1-form symmetries is the same as that of 0-form symmetries. Then, the standard reconstruction theorem should apply.

[4]This action has been discussed in recent literature [17–19] as we further discuss in the next section.

an effective pion electrodynamics in $d = 4$ with Lagrangian

$$\mathcal{L} = \frac{1}{2}\partial_\mu \pi_0 \partial^\mu \pi_0 - \frac{1}{4e^2}F_{\mu\nu}F^{\mu\nu} + \frac{1}{8\mu}\epsilon_{\mu\nu\rho\sigma}\pi_0 F^{\mu\nu}F^{\rho\sigma}, \tag{3}$$

where $\mu$ is a constant with dimensions of mass. The Euler-Lagrange equations read

$$\Box \pi_0 = \frac{1}{4\mu}\tilde{F}^{\mu\nu}F^{\mu\nu}, \tag{4}$$

$$\partial_\nu F^{\mu\nu} = \frac{e^2}{\mu}\tilde{F}^{\mu\nu}(\partial_\nu \pi_0). \tag{5}$$

The dual field-strength is defined by $\tilde{F}^{\mu\nu} \equiv \frac{1}{2}\epsilon^{\mu\nu\rho\sigma}F_{\rho\sigma}$. It is conserved $\partial_\nu \tilde{F}^{\mu\nu} = 0$. Notice that the first equation of motion (4), associated with the neutral pion field $\pi_0$, expresses the anomaly. Indeed we can rewrite this equation as

$$J^\mu = \mu \, \partial^\mu \pi_0, \qquad \partial_\mu J^\mu = \frac{1}{8}\epsilon_{\mu\nu\rho\sigma}F^{\mu\nu}F^{\rho\sigma}. \tag{6}$$

Following Adler, we can express (6) as a conservation equation of a gauge-dependent current

$$\tilde{J}^\mu = \mu \, \partial^\mu \pi_0 - \frac{1}{2}\tilde{F}^{\mu\nu}A_\nu, \qquad \partial_\mu \tilde{J}^\mu = 0. \tag{7}$$

This non-gauge invariant current can be integrated over a Cauchy slice to obtain a conserved charge

$$\tilde{Q} = \int d^3x \, \tilde{J}^0(x) = \int d^3x \left(\mu \, \dot{\pi}_0(x) - \frac{1}{2}B^i(x)A_i(x)\right), \tag{8}$$

where $B^i = -\frac{1}{2}\epsilon^{ijk}F_{jk}$ represents the magnetic field. This charge is now gauge invariant for gauge transformations or fields that vanish at infinity.

The question is then whether this generator gives rise to a symmetry of the theory in the conventional sense, and what are the differences, if any, with ordinary internal symmetries. A generic self-adjoint operator generates a group of unitaries but to be a generator of an internal symmetry other conditions must apply. More concretely, the group must leave algebras of local operators in themselves, and the transformations must commute with Poincare symmetries. Using canonical quantization we now check that this is the case in the present model.

Defining the electric field as $E^i = -F^{0i}$, the canonical momenta write

$$p_0 = \frac{\delta\mathcal{L}}{\delta\dot{\pi}_0} = \dot{\pi}_0, \qquad p_A^i = \frac{\delta\mathcal{L}}{\delta\dot{A}_i} = \frac{1}{e^2}E^i + \frac{1}{\mu}\pi_0 B^i. \tag{9}$$

We see that the pion momentum is unaffected by the anomalous term, while the conjugate momentum of the photon field gets a contribution. The non zero equal-time canonical commutation relations write

$$\left[\pi_0(x), p_0(y)\right] = i\,\delta(x-y), \qquad \left[A_i(x), p_A^j(y)\right] = i\,g_i^j\,\delta(x-y). \tag{10}$$

It is enlightening to see the implications of these canonical commutation relations on commutators of observables. The non trivial ones read

$$\left[B^i(x), E^j(y)\right] = i\,e^2\,\epsilon^{ijk}\,\partial_k^x\,\delta(x-y), \tag{11}$$

$$\left[p_0(x), E^i(y)\right] = \frac{i\,e^2}{\mu}B^i(y)\,\delta(x-y), \tag{12}$$

$$\left[E^i(x), E^j(y)\right] = -\frac{i\,e^4}{\mu}\epsilon_{ijk}\Big(\pi_0(y)\,\partial_y^k\delta(y-x) + \pi_0(x)\,\partial_x^k\delta(x-y)\Big). \tag{13}$$

The relations (12) and (13) show the effect of the interaction on the physical phase space of the theory as a deformation of the canonical commutation relations. Specially interesting is the non-commutativity of electric fields. These commutators play the role of the Schwinger terms found for QED in [35] and discussed below. In the pion effective theory, they arise by straightforward canonical quantization.

Using these commutators we can find the action of the charge (8) on the local field operators

$$\left[\tilde{Q}, \pi_0(x)\right] = -\mu\,i\,, \qquad \left[\tilde{Q}, B_i(x)\right] = 0\,, \qquad \left[\tilde{Q}, p_0(x)\right] = 0\,, \qquad \left[\tilde{Q}, E_i(x)\right] = 0\,. \tag{14}$$

The only non-vanishing commutator is with the pion field $\pi_0$ itself. This is as expected since all other fields do not carry a chiral charge. The symmetry is spontaneously broken. The pion is a Goldstone boson for the chiral symmetry and transforms additively

$$U(\lambda)\,\pi_0(x)\,U(\lambda)^\dagger = \pi_0(x) + \lambda\,\mu\,, \qquad U(\lambda) = e^{i\lambda\tilde{Q}}\,. \tag{15}$$

It is easily checked that the transformation implemented by $U(\lambda)$ respects the equations of motion and commutation relations.

We can move forward and compute the stress tensor. It reads

$$T^{\mu\nu} = \left(\partial^\mu\pi_0\partial^\nu\pi_0 - \frac{g_{\mu\nu}}{2}\partial_\alpha\pi_0\partial^\alpha\pi_0\right) + \frac{1}{e^2}\left(F^{\mu\alpha}F_\alpha{}^\nu + \frac{g_{\mu\nu}}{4}F_{\alpha\beta}F^{\alpha\beta}\right)\,. \tag{16}$$

A priori, there seems to be a tension between the canonical stress tensor, and the one obtained by deriving the action with respect to the metric in this model. This is because the latter, eq. (16), coincides with the free one. Nevertheless, it can be shown that a proper improvement exists and that (16) implements the correct time evolution. This is due to the non trivial commutation relations (12) and (13). We review these subtleties in appendix A.

With the previous commutators, we can check that the stress tensor is invariant under the modified chiral transformations, namely

$$\left[\tilde{Q}, T^{\mu\nu}\right] = 0\,. \tag{17}$$

This shows, as far as the effective model is concerned, that $\tilde{Q}$ generates a true internal symmetry of the theory. Moreover, it is not even necessary to think that this symmetry is implemented by a global unitary, since the transformation of the pion (15) is an automorphism of the local operator algebras and equations of motion. In particular, this implies one can find local transformations for any compact subregion of any topology, following [7–11], as we review below.

Having established the modified chiral symmetry as a $U(1)$ global symmetry, we now need to look for its implications. The first question concerns the sense in which this symmetry is different from other, more conventional, symmetries. In particular, there should be a reason why this symmetry does not have a Noether current since the current (7) is non gauge invariant. The reason turns out to be a particular case of the theorem shown in [11], because the present theory possesses HDV classes associated to regions with the topology of non contractible loops, that we promptly describe, and the symmetry does change these non local classes. However, it is important to realize that the description of this phenomenon does not involve more information than the one already given, and the action of the symmetry is completely specified by (14). In particular, we do not need to study the theory in non trivial manifolds to arrive at these conclusions and all commutation relations with non local operators follow from the commutation relations with the local ones.

To analyze the action of the chiral symmetry on the HDV classes let's further analyze the equation of motion (5) of the gauge field. These can be rewritten as the conservation of a gauge-invariant two-form field, namely[5]

$$G^{\mu\nu} \equiv \frac{1}{e^2} F^{\mu\nu} - \frac{\pi_0}{\mu} \tilde{F}^{\mu\nu}, \qquad \partial_\nu G^{\mu\nu} = 0. \tag{18}$$

The other gauge-invariant conserved two-form is the dual of the field strength itself

$$\partial_\nu \tilde{F}^{\mu\nu} = 0. \tag{19}$$

This allows us to define the corresponding conserved fluxes by integrating these currents over two dimensional oriented surfaces $\Sigma$ as

$$\Phi_G = \int_\Sigma \star G, \qquad \Phi_F = \int_{\tilde{\Sigma}} \star \tilde{F}. \tag{20}$$

To get non-trivial operators we need to integrate these fluxes over open $\Sigma$ with boundary $\partial \Sigma$ given by a loop. Because of the conservation of the fluxes, the resulting operators then commute with all local field operators spatially separated from $\partial \Sigma$ and in this sense, they can be considered loop operators. Consider these fluxes over two-dimensional surfaces at the $t = 0$ time slice

$$\Phi_G = -\int_\Sigma ds_i\, p_A^i = -\int_\Sigma ds_i \left( \frac{1}{e^2} E^i + \frac{1}{\mu} \pi_0 B^i \right), \qquad \Phi_F = \int_{\tilde{\Sigma}} ds_i\, B^i. \tag{21}$$

The commutator between these fluxes, when defined over two different surfaces with associated loop boundaries, is proportional to the linking number between such boundaries. The fact that this is a topological invariant follows from the commutativity of these fluxes with local operators outside, which allow to deform the loops without changing the commutator. From a direct calculation using (10-13) we obtain for simply linked loops (see [37, 38] for these concrete calculations)

$$[\Phi_G, \Phi_F] = i. \tag{22}$$

Of course, the same holds for the free electromagnetic field. In this effective model, the electric flux just gets modified (dressed with the chiral field) in order to be a loop operator.

Considered then a ring-like region $R$ with the topology of $D \times S^1$, where $D$ is a disk. The complementary region $R'$ in $\mathbb{R}^3$ has the same type of topology containing non-contractible loops. We can place both electric and magnetic type of loop operators with boundary $\partial \Sigma \in R$, and the same for $R'$. These operators with boundaries in $R$ cannot be constructed from the algebra of local operators on $R$. If that were the case, it would commute with the loop operators in $R'$, which is not the case. Then, they are not locally constructible in $R$ but commute will all operators formed by the algebra of local field operators spatially separated from $R$. Therefore, they violate Haag duality in $R$. Haag duality is violated for a region $R$ when the algebra of local operators on $R$ is smaller than the algebra of all operators that commute with the local operators on $R'$. In this sense, we call them non-local operators in $R$. However, they can be constructed locally as fluxes when considered inside topologically trivial regions. The notion of non-local is a relative one. Equivalently, all operators in this theory can be ultimately constructed from the local fields. This Haag duality violation is a genuine physical feature of the structure of the theory that is manifested in the way operator algebras live in different regions. See section 3.2 and [12, 13] for more details.

---

[5]This current was also identified in Ref. [36] but dismissed because of the compactness of the pion field. However, in this low energy effective theory, we should interpret the pion field as non-compact (since $\pi_0 \ll \mu$) leading (through a surface flux) to a well-defined HDV operator in the IR.

We now define the unitary operators, the Wilson loops (WL) and 't Hooft loops (TL), by exponentiating the (appropriately smeared) fluxes

$$W_q = e^{iq\Phi_F}, \qquad T_g = e^{ig\Phi_G}. \tag{23}$$

These are non-local operators associated with rings. The fusion rules for these non-local operators read

$$W_q W_{q'} = W_{q+q'}, \qquad T_g T_{g'} = T_{g+g'}. \tag{24}$$

In this effective model, the non-local operators of a ring like region $R$ form a $\mathbb{R} \times \mathbb{R}$ group.[6] A generic non local operator will be "dyonic", i.e., will have both electric and magnetic charges, and we will call it $D_{(g,q)}$. This is formed for example by products of WL and TL with charges $q$, $g$, in the same $R$. These operators in $R$ form classes invariant under the action of the local operators in $R$, and determined exclusively by the charges $(g,q)$. Equivalently, multiplying a given dyon by local operators in $R$ cannot change its class. The dual group, corresponding to $R'$, is also $\mathbb{R} \times \mathbb{R}$. The commutation relations for linked dyonic operators just follow from the ones of the fluxes themselves. They read

$$D^R_{(g,q)} D^{R'}_{(g',q')} = e^{i(q g' - q' g)} D^{R'}_{(g',q')} D^R_{(g,q)}. \tag{25}$$

As could have been anticipated, the action of the correct chiral charge $\tilde{Q}$ is particularly interesting on these operators. It transforms the electric fluxes while leaving the magnetic fluxes invariant. More precisely

$$[\tilde{Q}, \Phi_G] = i\Phi_F, \qquad [\tilde{Q}, \Phi_F] = 0. \tag{26}$$

Considering finite transformations on the loop operators one obtains

$$U(\lambda)W_q U^{-1}(\lambda) = W_q, \qquad U(\lambda)T_g U^{-1}(\lambda) = D_{(g,\lambda g)}. \tag{27}$$

The precise 't Hooft loop operator on the left hand side of the last formula depends on the precise form of the TL on the right hand side. The important point is the class to which it belongs. In words, 't Hooft loops (HDV magnetic classes) are charged under the action of the symmetry generated by $\tilde{Q}$. They mix with magnetic fluxes. For a generic HDV class, the transformation becomes

$$U(\lambda)D_{(g,q)}U^{-1}(\lambda) = D_{(g,q+\lambda g)}. \tag{28}$$

Because the symmetry keeps the algebra of local operators in $R'$ into itself, it will also keep invariant the set of all operators that commute with it. Then, it will map the algebra of local and non local operators in $R$ into itself, a fact that will appear again below. Though the classes are non-invariant under the modified chiral symmetry, the fusion rules and the commutation relations are invariant, as must be the case for an automorphism of the algebra:

$$D^R_{(g_1,q_1+\lambda g_1)} D^R_{(g_2,q_2+\lambda g_2)} = D^R_{(g_1+g_2,q_1+q_2+\lambda(g_1+g_2))}, \tag{29}$$

$$D^R_{(g,q+\lambda g)} D^{R'}_{(g',q'+\lambda g')} = e^{i(q g' - q' g)} D^{R'}_{(g',q'+\lambda g')} D_{(g,q+\lambda g)}. \tag{30}$$

The interpretation of this action is simple and transparent. We have an ordinary internal unitary symmetry that transforms any algebra of local operators for any region into itself. This action transforms the non local HDV classes. This is the new feature that is not generally found for other internal symmetries.

---

[6]Adding the massless charged pions would convert this to an $\mathbb{Z} \times U(1)$ group. This is consistent with the rest of this section provided we keep the symmetry transformations in the range of the effective model $\pi_0 \ll \mu$.

To end this section we find it important to remark two key facts that usually lead to confusion. First, both electric and magnetic non local operators form part of the ordinary algebra of local operators in balls, and as such they cannot be eliminated or excluded from the theory. Second, the behavior of the non local operators under the symmetry is, for the same reason, determined by the action of the symmetry on the local field operators. What the non local operators of the theory are is computed from the local operator algebras themselves. It is not possible to have a symmetry acting differently on local vs non-local operators in the QFT. In the present example, this is explicit since all commutators are determined from the canonical (local) ones. The interesting feature here is that the fluxes of the electric field are mixed with the chiral local field (the pion) in order to produce a non local operator in a ring. The electric flux by itself, which is non local for the free Maxwell field, is not a non local operator in a ring-like region $R$ in the present model because it does not commute with local operators outside $R$.

## 2.2 Origin of anomaly quantization

The effective pion model discussed above is non renormalizable and has to be completed in the UV. This completion will necessarily need the introduction of charges that break the HDV sectors from $\mathbb{R} \times \mathbb{R}$ to a smaller group. A more formal reason for this reduction of the sectors is that for the case of non compact sectors, the dual conserved two-form fields $F, G$ generating them must have a cross correlator that is fixed by the symmetry and does not renormalize. This leads to a free model [39].

In the completion of the model inside QCD with charged massless quarks, the HDV sectors are reduced to $U(1) \times \mathbb{Z}$, with a $U(1)$ of Wilson loops and discrete 't Hooft loops forming a group $\mathbb{Z}$.[7] In the effective model this could be incorporated by introducing the charged pions. Let us consider such a compactification from a more general perspective. We set the minimal electric charge to $q = q_0$, where the covariant derivative is $\partial_\mu + iqA_\mu$. This corresponds to setting to $eq_0$ the corresponding Coulomb charge in the weakly coupled electromagnetic field. Then, for the WL we have a compact $U(1)$ group labeled by the charges $q \in [0, q_0)$. In consequence the TL form a group $\mathbb{Z}$ with charges $g = \frac{2\pi}{q_0} k$, and $k$ integer. Crucially, the compatibility of the compact group of WL with the non trivial action of the chiral symmetry on the $U(1) \times \mathbb{Z}$ group of HDV sectors implies restrictions for the radius of compactness of this symmetry. This is the first important consequence of the $U(1)$ symmetry that we now describe.

To this end, we first notice that any extension of the symmetry to higher energies must respect eq. (28). This implies that the range of the parameter $\lambda$ of the chiral $U(1)$ symmetry is

$$\lambda \in [0, \lambda_0), \qquad \lambda_0 = n \frac{q_0^2}{2\pi}, \tag{31}$$

where $n$ is a positive integer. In terms of the model with Lagrangian (3) this equation relates the radius of compactification of the pion field with the coefficient of the anomalous term. The conventional definition of the pion decay constant gives the pion compactification radius as $\pi_0 \equiv \pi_0 + 2\pi f_\pi$. Using eq. (15) for the generator of the chiral symmetry and (31) we get the coefficient of the anomalous term as

$$\frac{1}{8\mu} = \frac{n q_0^2}{32\pi^2 f_\pi}. \tag{32}$$

---

[7]Both the non-compact $\mathbb{R} \times \mathbb{R}$ and compact $U(1) \times \mathbb{Z}$ groups of HDV classes furnish examples of the principle that generalized symmetries come in dual pairs [13], see [40] for a recent non-trivial application of this principle in the context of general linearized gravity theories.

This equation expresses the quantization of the value of the anomaly coefficient in purely group theoretical terms, as the necessary compatibility between the HDV classes and a global symmetry that acts non trivially on them.

The quantization of the anomaly has been obtained previously from different considerations. In the immersion of this model in a non linear sigma model, appropriate for QCD with massless quarks and a number of flavors $N_f \geq 3$, the quantization of the coefficient follows from the quantization of the WZW coefficient by topological reasons [41].[8] In this case we have $q_0 = 1/3$ and $n = 3N_c$ with $N_c$ the number of colors [42].

Another way to express these features is in terms of a current $\hat{J} = \frac{\lambda_0}{2\pi} J$ normalized such that the charge operator has a cycle set to the standard value $2\pi$. This implies the charge has integer eigenvalues. From (6) this chiral current has the anomaly

$$\partial_\mu \hat{J}^\mu = n \frac{q_0^2}{32\,\pi^2} \, \epsilon_{\mu\nu\rho\sigma} F^{\mu\nu} F^{\rho\sigma} \,. \tag{33}$$

We see this formula conforms well with the general value of the anomaly. For the chiral current in QED we have $n = 2$ if the chiral angle is the phase acting on the electron, as corresponds to having two chiral fields for the Dirac fermion. But we have $n = 1$ if we set the minimal chiral charge on gauge invariant operators to 1 (see below).

For $n > 1$ the chiral symmetry is moving $n$ times faster in the non local classes than in the pion field. Operators with unit chiral charge are obtained by exponentiating smeared pion fields:

$$e^{i f_\pi^{-1} \int \alpha(x)\pi_0(x)}, \qquad \int \alpha(x) = 1 \,. \tag{34}$$

On the other hand, we can produce chirally charged operators by combining non local operators as

$$\Psi_{k,m} = \int_0^{q_0} dq \, e^{i m \frac{2\pi q}{q_0}} D_{\left(\frac{k 2\pi}{q_0}, q\right)} \,, \tag{35}$$

for integer $m$.[9] The chiral transformation gives

$$U(\lambda) \Psi_{k,m} U(\lambda)^\dagger = e^{i n m k \frac{2\pi\lambda}{\lambda_0}} \Psi_{k,m} \,. \tag{36}$$

The minimal non zero charge with respect to the chiral symmetry in these type of operators is then $n$, attained for $m = 1$ and the elementary TL with $k = 1$. This provides a complementary physical interpetation of the integer $n$ that defines the anomaly.

## 2.3 Chiral symmetry in massless QED

We consider now the case of massless QED in $d = 4$. This theory is described by the action

$$S = \int d^4x \left[ -\frac{1}{4e^2} F_{\mu\nu} F^{\mu\nu} + \overline{\psi}\, i\slashed{\partial}\, \psi - \overline{\psi}\slashed{A}\psi \right] \,. \tag{37}$$

In this case, the chiral symmetry under transformations of the form $\psi \to e^{-i\alpha\gamma^5}\psi$ is associated by Noether's theorem with the current

$$J_5^\mu = \overline{\psi}\, \gamma^\mu \gamma^5 \, \psi \,. \tag{38}$$

---

[8]Notice our derivation of the quantization of the anomaly does not require any assumption on the number of flavors.

[9]Note that selecting special operator representatives of the non local classes that transform into themselves by the group operation as in (28) automatically eliminates additional charges from local operators. The construction of these operators can be accomplished in a standard way using modular tools, see [12], section 2.2.3.

However, the associated conservation law is anomalous at a quantum level [1,2]. This anomaly reads

$$\partial_\mu J_5^\mu = \frac{1}{16\pi^2} \epsilon^{\mu\nu\rho\sigma} F_{\mu\nu} F_{\rho\sigma} \,. \tag{39}$$

As in the pion example, we can define a conserved but non-gauge invariant current by means of

$$\tilde{J}_5^\mu = J_5^\mu - \frac{1}{4\pi^2} \tilde{F}^{\mu\nu} A_\nu \,, \qquad \partial_\mu \tilde{J}_5^\mu = 0 \,. \tag{40}$$

This gives us a gauge invariant conserved global charge operator when integrated over all the space

$$\tilde{Q} = \int d^3 x \left[ \psi^\dagger(x) \gamma^5 \psi(x) - \frac{B^i(x) A_i(x)}{4\pi^2} \right]. \tag{41}$$

To understand whether this charge defines an internal symmetry or not we need to know how it acts on local fields. The canonical momenta can be computed to be

$$p_\psi = i\psi^\dagger \,, \qquad p_A^i(x) = \frac{1}{e^2} E^i \,, \tag{42}$$

implying the equal-time (anti)commutation relations

$$\left\{ \psi(x), \psi^\dagger(y) \right\} = \delta(x-y) \mathbb{1} \,, \qquad \left[ A_i(x), E^j(y) \right] = ie^2 g_i^j \delta(x-y) \,. \tag{43}$$

From here, one would naively think the action of the charge (41) is given by

$$\left[ \tilde{Q}, \psi(x) \right] = -\gamma^5 \psi(x) \,, \qquad \left[ \tilde{Q}, \psi^\dagger(x) \right] = \psi^\dagger(x) \gamma^5 \,, \tag{44}$$

$$\left[ \tilde{Q}, A_i(x) \right] = 0 \,, \qquad \left[ \tilde{Q}, E^i(x) \right] = -\frac{i}{2\pi^2} B^i(x) \,. \tag{45}$$

This acts as expected on the fermion field by swapping its chiralities. However, it acts on the photon field in a non-covariant manner by changing the electric field into a magnetic field and leaving the magnetic field invariant. This is not only strange but also inconsistent with the fact that $\dot{\tilde{Q}} = 0$. This issue is resolved by considering the Schwinger terms appearing in the commutator with the composite charge density. The non vanishing Schwinger terms[10] at leading order in perturbation theory are given by [35]

$$\left[ J_5^0(x), E^i(y) \right] = \frac{ie^2}{2\pi^2} B^i(x) \delta(x-y) \,, \tag{46}$$

$$\left[ J_5^i(x), E^j(y) \right] = \frac{ie^2}{4\pi^2} \epsilon^{ijk} E_k(x) \delta(x-y) \,, \tag{47}$$

$$\left[ J^0(x), J_5^0(y) \right] = \frac{i}{2\pi^2} B^i(y) \partial_i^x \delta(x-y) \,, \tag{48}$$

$$\left[ J^i(x), J_5^0(y) \right] = \frac{i}{2\pi^2} \epsilon^{ijk} E_j(x) \partial_k^x \delta(x-y) \,, \tag{49}$$

$$\left[ J^0(x), J_5^i(y) \right] = -\frac{i}{2\pi^2} \epsilon^{ijk} E_j(x) \partial_k^x \delta(x-y) \,. \tag{50}$$

In consequence (45) is modified as

$$\left[ \tilde{Q}, A_i(x) \right] = 0 \,, \qquad \left[ \tilde{Q}, E^i(x) \right] = 0 \,. \tag{51}$$

---

[10]One may inquire for the possible existence of further Schwinger terms. However, as computed in [35], the full set (50) is consistent with all the equations of motion and conservation laws of the theory. This, combined with the fact that on general grounds one would not expect more derivatives of the delta functions in (50) [35, 43], suggests that (50) is indeed exact to all orders in perturbation theory.

It is clear that $\tilde{Q}$ leaves the photon variables unchanged as we would have expected from the previous example. The smallest chiral charges are given by chiral bilinears

$$\left[\tilde{Q}, \bar{\psi}\left(\frac{1 \pm \gamma^5}{2}\right)\psi(x)\right] = \pm 2\bar{\psi}(x)\left(\frac{1 \pm \gamma^5}{2}\right)\psi(x). \tag{52}$$

These bilinears have two units of charge as mentioned above. By similar computations, we can also check that the modified charge, which implements a rather standard chiral transformation, does commute with the stress tensor. Notice this computation again requires the consideration of the previous Schwinger terms.

We conclude that the transformations generated by the charge (41) do seem to obey all the requirements of internal symmetry. In particular, since it transforms local algebras into themselves and commutes with spacetime symmetries, we can construct local charges (or local twists) for any subregion of any given topology, following [7–11], as we review below.

As before, the remaining question is in which sense, if any, this symmetry is different from conventional internal symmetries. To approach this question we notice this theory has HDV sectors for ring-like regions, given by a $\mathbb{Z} \times U(1)$ group. The $\mathbb{Z}$ part corresponds to TL with charges $2\pi k$, while the $U(1)$ part corresponds to WL of charges in $q \in [0, 1)$. Wilson loops with integer charges are not non-local operators because they are locally decomposable into Wilson lines. As $\tilde{F}$ is still conserved the WL can be constructed exponentiating fluxes of the magnetic field. The chiral symmetry leaves the WL classes invariant, because the magnetic field is invariant. Only TL of charges $2\pi k$ are loop operators. This, and the commutation relations with WL, is fixed by the non local WL classes forming a $U(1)$ group.

The crucial question now asks for the nature of the TL. We notice first that these TL operators necessarily form part of the algebra generated by local operators in a ball.[11] The reason is that the WL belongs to this algebra (it is a flux of the magnetic field), and von Neumann's double commutant theorem requires the TL to exist in the algebra [13]. An expression of the TL showing that it belongs to the algebra of local operators could be obtained in an abstract way using modular theory. But it is certainly obscure how to make this construction explicit in this QED context. The previous pion model gives a hint on what is going on in the expression of the TL in terms of local operators. Certainly, chirally charged local operators must play a role. The standard way to express the action of the TL is the original definition by 't Hooft as an insertion of a boundary condition along $\Gamma = \partial\Sigma$ in the path integral [44]. This boundary condition is the imposition of a magnetic monopole-like condition on the gauge field on a $S^2$ sphere around any point in $\Gamma$. This insertion necessarily has the right commutation relations with WL, a fact that establishes the TL as a HDV operator in the ring.

We now want to analyze the chiral transformation of the TL. This transformation can be computed from the Witten effect [34], and in this way, it directly connects with the same effect on the pion electrodynamics. Witten's effect describes a monopole of magnetic charge $g$ subject to an external change of the theta term parameter in the Lagrangian

$$L_\theta = \frac{1}{16\pi^2}\, \theta\, F^{\mu\nu}\tilde{F}_{\mu\nu}, \tag{53}$$

for a total change $\Delta\theta$. In this process a monopole with charge $g$ is transformed into a dyon of charge

$$\left(g, g\,\frac{\Delta\theta}{4\pi^2}\right). \tag{54}$$

Equivalently, the monopole has acquired an electric charge $g\,\frac{\Delta\theta}{4\pi^2}$. As a result, monopole boundary conditions along the TL are also modified to dyon boundary conditions, and the change of the TL to dyon loop is given by the same formula (54). For more details see for example [42].

---

[11]Such ball must fully contain the ring-like region that encloses the loop on which the TL is defined.

Now it only remains to connect such change of $\theta$ with a chiral transformation through the anomaly. In effect, such term appears in the action as a result of a chiral transformation of the fermion measure in the path integral [45]. The chiral transformation changes the action with the term 53, where

$$\Delta \theta = \left( \sum_i q_i^2 \right) \lambda \,. \tag{55}$$

In this equation, $\lambda$ is precisely the angle of the chiral transformation, and the sum is over the different chiral fermions with charge $q_i$. For QED this is $\Delta \theta = 2\lambda$. Notice we could also have invoked Witten's effect in pion electrodynamics, with the identification

$$\theta = n q_0^2 \frac{\pi_0}{f_\pi} \,. \tag{56}$$

Therefore in QED we have the same phenomenon of a chiral $U(1)$ symetry that changes the classes as

$$(2\pi k, q) \rightarrow \left( 2\pi k, q + 2k \frac{\lambda}{2\pi} \right) , \tag{57}$$

with $\lambda \in [0, 2\pi)$, $q \in [0, 1)$. We see the minimal charge constructed with these loops is 2, but this is also the minimal charge in the gauge invariant local operators. In this sense, this model achieves the minimal possible value $n = 1$. Analogous descriptions of the non trivial action of the chiral transformation on the TL have appeared previously [17–19]. However, the interpretation and consequences differ. We will compare with such works in section 3.

It is interesting to note that in QED this symmetry is a priori expected to be unbroken. In fact, ordinary QED does not present any Goldstone scalar and the electron mass is already very small with respect to the scale at which the coupling constant is strong. This would lead to the equation $\langle D_{(g,q)} \rangle = \langle D_{(g,q+\lambda g)} \rangle$ for symmetry related non local operators. This equation is certainly surprising from the point of view of the free Maxwell field. One may wonder how this is compatible with the "wrong" sign of the beta funtion in QED. It would be interesting to have a more explicit understanding of this equation.

## 2.4 Anomaly matching and Goldstone theorem

The usual understanding of anomalies is that they do not renormalize. Equivalently, the divergence of the anomalous current must be the same in the UV and IR models. This implies that the generic expression for the anomaly

$$\partial_\mu J_5^\mu = c \, \epsilon^{\mu\nu\rho\sigma} F_{\mu\nu} F_{\rho\sigma} \,, \tag{58}$$

has the same dimensionless $c$ for the UV and IR theories, when the normalization of the current and electromagnetic field are fixed. This enforces an anomaly matching between the IR and UV models. This matching famously implies the existence of massless excitations in the IR, either fermions or Goldstone bosons, capable of reproducing the UV anomaly [16]. For example, for QCD with massless quarks we have for the anomaly in the chiral current

$$J^\mu = \frac{1}{2} \left( \bar{u} \gamma^\mu \gamma^5 u - \bar{d} \gamma^\mu \gamma^5 d \right) , \tag{59}$$

the value

$$c = \frac{N_c}{2} \frac{1}{16\pi^2} \left( \left( \frac{2}{3} \right)^2 - \left( \frac{1}{3} \right)^2 \right) = \frac{N_c}{96\pi^2} \,. \tag{60}$$

This follows from the QED anomaly for charges $2/3$ and $-1/3$ for the quarks $u$ and $d$, which are replicated by the number of colors $N_c$. The normalization of $F$ is fixed by setting the charge

of the proton to one. This current is identified at the IR with the one creating the pion field $J_\mu = f_\pi \, \partial_\mu \pi_0$. Anomaly matching then gives us the coefficient on the pion Lagrangian through eqs. (32, 33), with $q_0 = 1/3$, $n = 3N_c$, as we have already mentioned.

We now want to understand anomaly matching in terms of the existence of the global $U(1)$ chiral symmetry. Notice that, generically, the existence of an ordinary continuous global symmetry in the UV does not necessarily entail massless excitations in the IR. Although massless Goldstone bosons appear if the symmetry is spontaneously broken, it can be the case that all charged particles become massive, and the symmetry effectively disappears in the IR. However, for the present chiral symmetry, the key input is that it changes the HDV classes. Therefore the symmetry cannot just disappear in the IR since it transforms the non local classes of the electromagnetic field. These classes are also present at the IR since it is assumed the electromagnetic field becomes weakly coupled in that regime. Then it cannot be the case that a symmetry changes classes at the UV and then stops changing them at the IR, if the classes are still present. This is not possible because the action of the symmetry on the non local classes is preserved under "transportability", or deformations of the non local operators by local ones [11]. These deformations can continuously transform a small loop to a large one. Another perspective is the following. The one form symmetry does not disappear in the IR because of the existence of the Maxwell field. Then the TL have non trivial expectation values, giving place to non trivial chiral charged operators. Quantum complementarity forces the existence of dual operators to these charged ones, with non trivial expectation values as well [13, 46]. In this case, these are the local twist effecting the symmetry transformation in the IR.

On the other hand, as we have remarked before, the non local operators are constructed with ordinary local operators. Since TL are chirally charged, this implies that in the IR there must be local operators that are still transforming under the symmetry. This accounts for the existence of massless excitations at the IR (besides the photon field). Then, anomaly matching becomes simply the statement that the action of the group symmetry on the non local classes is preserved across the RG flow, and therefore its manifestations can be matched across scales. It is clear that the velocity of this transformations on the preserved classes also has to be preserved, and this velocity sets the coefficient of the anomaly.

In this vein it could be the case that not all the non local sectors are preserved, or, as in QCD, new sectors emerge at the IR because the minimal charge changes from $1/3$ to $1$ (considering the charged pions). In this case, the UV sectors form a subgroup $U(1)_{\mathrm{IR}}/Z_3 \times 3\mathbb{Z}_{\mathrm{IR}}$ of the infrared ones. Matching the action of the chiral symmetry then leads to

$$n_{\mathrm{UV}} \, (q_0^{\mathrm{UV}})^2 = n_{\mathrm{IR}} \, (q_0^{\mathrm{IR}})^2 \,. \tag{61}$$

Finally, having a continuous global symmetry in the QFT implies that Goldstone theorem applies [47–49]. The validity of this theorem does not rest in the existence of a gauge invariant current. It only requires the existence of the gauge invariant local charges or twists, as proven in [50]. As we discuss in the next section, local $U(1)$ charges can be constructed for the pion field. Therefore, this symmetry can be spontaneously broken or not in the IR, leading to the existence of Goldstone modes or not. In QCD the first possibility arises, with the appearance of the pions as the Goldstone bosons of the chiral symmetry breaking. In other words, the coupling between QCD and the electromagnetic field, which spoils the naive conservation of the chiral current, does not spoil the applicability of Goldstone's theorem, implying the existence of neutral pion at low energies.[12]

---

[12]A version of Goldstone theorem for non-invertible symmetries explaining these aspects was described in [21]. But from the present analysis, it is clear it is the conventional Goldstone theorem the one dictating the pion field is a Nambu-Goldstone boson.

## 2.5 Local implementation of the symmetry and Noether's theorem

Given an internal symmetry, defined as a global automorphism of the locally generated (additive) algebras for any spacetime region, there are always operators supported in compact regions of the space that implement the symmetry inside these regions, acting trivially outside. This is the content of the weak form of Noether's theorem [7–10]. These operators implementing local transformations are called twists.[13] They are constructed in a general and abstract way using the split property and modular theory. The input are two commuting algebras with no non trivial intersection between them, and a global unitary effecting automorphisms for those algebras. The output are the twists themselves, effecting the transformation in one of the algebras. In the application to QFT, one chooses say a compact region $R$ and the complement $R'$, with a smearing zone $Z$ in between them. The compact region can have any topology. This construction was applied to QFT's displaying generalized symmetries in [11], where different choices for the two commuting algebras appear for each choice of region $R$ and its complement $R'$. These different choices of algebras lead to twists with different properties, as we review below.

The weak version of Noether's theorem and the construction of local twists can be extended to the symmetry breaking case [50]. This extension is relevant for the present purposes, namely for QCD with massless quarks, since the chiral symmetry is clearly spontaneously broken. In this case, the automorphisms of the local algebras commuting with Poincare symmetry replace the global symmetry operators (that are ill-defined in the symmetry-breaking case) in the construction of the twists. In general, there are many ways to construct the twists, differing on details on the boundary of the region.[14] In the models of the preceding section twists for any topology of the region $R$ can also be constructed in a more pedestrian way. This involves integrating the gauge non invariant charge density over $R$ and dressing it appropriately at the boundary of the region [11] (see also [17]).

Therefore, the existence of these local implementations allows us to study, for example, the transformations of the HDV classes under the symmetry, where now all symmetry transformations are locally defined and represented by the twist operators $\tau_g$. If the model is defined in a space of non trivial topology, the structure of the local algebras inside a contractible ball, for example, must be in principle the same, namely the additive algebra (see [51] for a recent discussion with different motivations). Twists for such balls will exist, and in turn from them we can provide twists associated with any given region (of any topology) inside the ball. This gives us an understanding that the chiral symmetry is the same invertible $U(1)$ symmetry in any spacetime, as far as the automorphisms of the local algebras are concerned. This symmetry is intrinsic to the QFT and, most importantly, it controls pion decay. We will discuss the global symmetries for compact manifolds with non-trivial topology in the next section, although as we will see, the possible features of these global operators have nothing to do with pion decay.

When the global symmetry changes the non local classes associated with a certain region, there are certain refinements in the classification of twist operators. More precisely, local twists for regions containing non local operators can reproduce the action of the global symmetry on these non local operators (complete twists) or not. More generally twists can implement the symmetry in a closed subalgebra of the non local operators that is kept invariant under the symmetry. Also, the local twists can belong to the local algebra of the region (additive twists),

---

[13]Conventional global symmetries are sometimes defined as those with topological operators for any codimension one manifold in Euclidean signature. Notice the present weak version of Noether's theorem in QFT shows one only needs to demonstrate the existence of the global topological operator in the manifold $M$ of interest, and that such global topological operator generates an automorphism of the additive algebras. Once these two things are demonstrated in the QFT at hand, the theorem unraveled in [7–10] implies the existence of topological operators for any subregion in manifold $M$.

[14]These ambiguities are sometimes hidden in the standard formulation of generalized symmetries. We will comment about them in section 3.

or not. All of these classes of twists can be constructed generally, with modular tools. They arise because when choosing the two commuting and non intersecting algebras, appropriate for the split construction, one has more than one choice. One can choose the additive algebras for both regions, or the additive algebra for one region, and the maximal algebra (including the non local operators) for the other. One can also choose intermediate choices, as long as they commute and do not intersect. The twists will form the same group as the global one and will act in the correct way on any operator whose transformation by the symmetry is kept in the algebra. In explicit examples, one can just dress the charge adequately on the boundary to form a twist. For more details on both approaches see [11].

It is immediate that a twist constructed with local operators in $R$, namely an additive twist, cannot change the non local classes, even if it acts correctly on local operators. This is simply because the twist is local in $R$ by assumption, and non local classes are defined by a quotient over the local algebra. In particular, a twist constructed with a (gauge invariant) Noether current is locally generated inside the region. Therefore it cannot change classes. Because local Noether charges concatenate, i.e., one can sum Noether charges for adjacent regions and obtain the one for the union of the regions, one can easily show that the Noether twist is also complete [11]. Therefore, it is not possible that the symmetry transforms the non local classes if there is a Noether current. This provides a transparent reason why in the present anomalous models it is not possible to construct a current for chiral symmetry.

## 2.6 Classifying non invariant classes under a $U(1)$ symmetry

We end this section by starting a classification, from a general point of view, of the possible structures in which we have HDV classes non invariant under a one parameter group. In this vein consider a one parameter group $G$ of symmetries with additive parameter $\lambda$. We assume this group acts non trivially on the HDV classes associated with a region $R$. More precisely we assume that the full group $G$ does not leave the classes invariant, and not that just a discrete subgroup moves the classes non trivially. Let us now call $a$ to a non local operator in $R$ that is not invariant. Then, there is necessarily a continuum of classes $a(\lambda)$.

In these type of scenarios, the simplest case arises when these $a(\lambda)$ are the only non local sectors. Then the fusion of $a$ is necessarily Abelian forming a one parameter group $A$, and we can label $a_1 + a_2$ the class generated by the product of sectors $a_1$, $a_2$, with $a = 0$ corresponding to the identity class. The action of the symmetry that respects this fusion algebra must be of the form

$$a(\lambda) = e^{\lambda} a, \tag{62}$$

where we have normalized the group parameter $\lambda$ such that the exponent does not have an extra constant factor. In the region $R'$ we have the dual non local operators $b$. The dual of the Abelian group $A$ is another Abelian group $B$ formed by the characters of $A$, and we can set the parametrization of the $b$'s such that the fusion of $b_1$ and $b_2$ is $b_1 + b_2$. The commutation relations between non local classes are fixed to be [12]

$$a\,b = e^{i\,a\,b}\,b\,a. \tag{63}$$

The only possible action of the symmetry on the dual sectors $b$ that respects this commutation relation is

$$b \to e^{-\lambda}\,b. \tag{64}$$

This gives a continuum of $b$, and the generalized symmetry is necessarily a $\mathbb{R}$ group for both $A$ and $B$. The symmetry group is a non compact $\mathbb{R}^+$ group. This is the case, for example, of the HDV sectors associated with the algebra of the derivatives of a free massless scalar field for $d \geq 3$ under the action of the dilatation group [11]. This is a free theory. When the sectors

of $R$ form a continuous non compact manifold (or some part of it is continuous non compact) we call the generalized symmetry non compact. In this case the same happens for the dual classes. We expect free theories when there are non compact sectors, and this was proven in the case that these non compact classes are generated by form fields [39].

Let us analyze next the case of HDV classes formed by an Abelian group $A$ with elements labeled $a = (a_1, a_2)$, where the fusion is additive in this vector parametrization. The coordinates can form a group $\mathbb{Z}$, $\mathbb{R}$, or $U(1)$. The dual group $B$ has elements $b = (b_1, b_2)$, with additive fusion. The commutation relations can be written

$$a \, b = b \, a \, e^{i a \cdot b} \,. \tag{65}$$

To respect these fusion rules and commutation relations we need an action of the symmetry of the form

$$a \to M(\lambda) a \,, \qquad b \to (M(\lambda)^T)^{-1} b \,, \tag{66}$$

where $M(\lambda)$ is a one parameter group of two dimensional real matrices.

The case $A = U(1) \times U(1)$, that gives $B = (\mathbb{Z}, \mathbb{Z})$, or viceversa, cannot have the action of a continuous symmetry. The reason is the discreteness of one of the dual sectors. Both dual sectors have to contain continuum parts for a non trivial action to be possible.

For the case $A = \mathbb{R} \times \mathbb{R}$, that has $B = \mathbb{R} \times \mathbb{R}$, the group can be any one parameter subgroup of $GL(2, \mathbb{R})$. This includes for example dilatations as the one discussed above, and a rotation. In this last case, the symmetry group is $U(1)$. An example of a rotation of HDV classes is given by the electromagnetic duality symmetry of the free Maxwell field in $d = 4$, or the rotation between two independent Maxwell fields. All these cases correspond to non compact generalized symmetries and are free. Another, effective field theory scenario, valid at the classical level, is the pion electrodynamics discussed above.

If $A = \mathbb{R} \times U(1)$, then $B = \mathbb{R} \times \mathbb{Z}$. We have $a_2 \equiv a_2 + 2\pi$ and $b_2 \in \mathbb{Z}$. The general symmetry is a combination of a dilatation in the non compact dual $\mathbb{R}$ sectors and the transformation

$$(a_1, a_2) \to (a_1, a_2 + \lambda a_1) \,, \qquad (b_1, b_2) \to (b_1 - \lambda b_2, b_2) \,. \tag{67}$$

We do not have examples of this type, though the non compactness of the sectors would imply a free model if they were generated by a form field [39], assuming the $\mathbb{R} \times \mathbb{R}$ part of the symmetry is not an effective but an exact one. In that case, it does not seem possible that this transformation can be realized.

Finally, we have the case $A = \mathbb{Z} \times U(1)$, with dual $B = U(1) \times \mathbb{Z}$. The only possible action of $G$ is the one given in (67). This is exactly the case of the chiral anomaly for $d = 4$ discussed in this paper, where the $\mathbb{Z}$ corresponds to TL and the $U(1)$ to WL. In this example the structure of the dual sectors in $R$ and $R'$ is the same because they have the same topology.[15] We do not know if this type of transformation is possible for regions $R$ and $R'$ of different topology. If the symmetry is a $U(1)$ the range of different $\lambda$ is $\lambda \in [0, 2\pi n)$, with $n$ an integer. This follows from the fact that $\lambda = 2\pi n$ has to act as the identity in (67), and we have set the periodicity of the $U(1)$ non local sectors to be $2\pi$.

The continuous non trivial symmetry implies both $R$ and $R'$ display some continuous HDV sectors, but this does not imply non-compactness since the continuous sectors can commute with each other, as in the previous example. These cases were not imagined in [11].

## 3 Comparison and discussion of previous literature

This article was mainly motivated by the interesting interplay between theories with ABJ anomalies, Noether's theorem and the conjecture we made in [11]. The previous discussion

---

[15]The different sign on $a_1$ and $b_2$ in the transformation (67) can be eliminated by redefining $b_2 \to -b_2$.

concluded that ABJ anomalies should be more properly understood as theories with a $U(1)$ global symmetry, as anticipated by Adler. The only features that make this global symmetry special is that it transforms the non-local classes. At once, this explains the quantization of the anomaly by the compatibility of $U(1)$ cycles, anomaly matching by symmetry preservation along the RG flow, the absence of a Noether current by the fact that local twists do transform the non local classes while Noether twists cannot, and the applicability of Goldstone theorem which shows the pion field can be considered a Goldstone boson even when QCD is coupled to the photon field.

These conclusions turn out to be in contrast with recent literature [17–19]. While part of the spirit and several computations are similar, we do not find that chiral symmetry is reduced to $Z_{n_f}$ in QED with $n_f$ massless fermions [17], or that it should be interpreted as non invertible [18, 19]. These discrepancies originate on overlooked features associated with the local physical manifestations of the generalized symmetries. The physics of QFT's with ABJ anomalies amplifies their consequences. The fact that these local features are overlooked is quite uniform in the generalized symmetry literature, as we have found in several discussions. We believe this originates in an unfortunate terminology that obscures these features. Since no local manifestations associated with the generalized symmetries are identified, the description of the phenomenon usually involves defining the theory in spaces with different topologies. For the models with ABJ anomaly, and many phenomena related to gauge theories, this seems to us both unnecessary and artificial. Putting the models in different manifolds necessarily involve arbitrary choices for the global structure of the algebra or Hamiltonian (superselection sectors, boundary conditions), while the local physics, that genuinely characterize the theory and its symmetries, can as well be studied in flat space. In flat space (as well as in other topologies) the essence of the phenomenon lies in the characterization of order/disorder parameters as operators violating Haag duality in particular regions. Equivalently, all order/disorder parameters are HDV operators and viceversa.

Therefore, as stated in the introduction, clarifying the overall physical picture in comparison to the previous works becomes a further important motivation that we develop in this section. We will start by explaining those features in general, then compare such features with the standard formulation of generalized global symmetries, and finally describe how they impact the understanding of the anomaly in the previous works.

## 3.1 Order/disorder parameters are HDV operators and viceversa

We will first briefly describe the classification of operators in QFT's with generalized symmetries that was put forward in [12, 13]. The essence of the classification lies in acknowledging and distinguishing the two different meanings that are usually assigned to the idea of locality. One sense of locality corresponds to the idea that an operator is formed by local degrees of freedom. For any region $R$ there is one intrinsic algebra associated with it, namely the additive algebra. Intuitively, this is the algebra generated by arbitrary products of (gauge invariant) local operators inside the region. It is the algebra an observer/laboratory in such a region would have access to. Formally, it can be self-consistently defined as

$$\mathcal{A}_{\mathrm{add}}(R) \equiv \bigvee_{B \, \mathrm{ball}, \cup B = R} \mathcal{A}(B). \tag{68}$$

The other sense, or idea, associated with locality is that operators associated to spatially separated regions commute. This is also called "causality". An immediate question asks if these two notions of locality end up being essentially one or not.

To approach this question we first notice a simple but basic observation. If the algebras associated with balls satisfy causality, the additive algebras for any given region, irrespective of its topology, satisfy causality by construction as well. However, for a region $R$ there may

be operators $a$ that are causal or local in the sense that they commute with all operators in $\mathcal{A}_{\text{add}}(R')$, where $R'$ is the causal complement of $R$, but that cannot be generated by local operators in $R$. We call non local operators in $R$ to this type of operators. Taking a complete set of these non local operators we have

$$(\mathcal{A}_{\text{add}}(R'))' = \mathcal{A}_{\text{add}}(R) \vee \{a\}, \tag{69}$$

where $\mathcal{A}'$ is the commutant of the algebra $\mathcal{A}$, and the symbol $\vee$ is used for the algebra generated by two sets of operators. In this situation, it is said that the additive net of algebras $\mathcal{A}_{\text{add}}(R)$, which by definition satisfies causality, does not satisfy Haag duality. Haag duality would be the property $\mathcal{A}_{\text{add}}(R) = (\mathcal{A}_{\text{add}}(R'))'$, and this is a precise form of completeness in the operator content of the theory [13]. Equivalently, the non locally generated operators $a$ violate Haag duality in the region $R$, and can be taken as the generators of the Haag duality violating (HDV) classes.[16] In these types of theories, the first sense of locality (related to whether operators can be generated by local fields or not) and the second sense (related to causality) differ in a physically meaningful sense.

Interestingly, when Haag duality is violated for $R$, eq.(69), it follows that the same is true for the complementary region $R'$, namely

$$(\mathcal{A}_{\text{add}}(R))' = \mathcal{A}_{\text{add}}(R') \vee \{b\}, \tag{70}$$

holds due to the existence of operators $b$ that are non local in $R'$. This is a consequence of von Newmann's double commutant theorem $\mathcal{A}'' = \mathcal{A}$. Then, the existence of dual sets of non local operators corresponding to complementary topologies is unavoidable, and dual non local operators enter in equal footing in the description. Another simple consequence is that all $a's$ cannot commute with all $b's$, otherwise there is Haag duality.[17] It is this non trivial action of the $a$ operators on the $b$ operators, and viceversa, that represents the generalized symmetry in this description. Examples of non local operators in ring-like regions are Wilson loops for uncharged representations, and for the complementary region, the 't Hooft loops that are not made additive by the existence of monopoles.

A simplifying assumption that holds quite generally is that the additive algebra, when considered for topologically trivial regions $B$ such as balls, satisfies Haag duality

$$\mathcal{A}_{\text{add}}(B) = \mathcal{A}_{\text{add}}(B')'. \tag{71}$$

Violations of this property are found to be unrelated to 1-form or higher form symmetries but appear for orbifolds of QFT's with a spontaneous breaking of a global symmetry. In that case, it can be simply repaired by taking the dual net [50], or, equivalently, considering the charged operators in addition to the neutral ones. Other examples correspond to QFT's that are pathological for other reasons, such as generalized free fields that do not have a stress tensor nor a causal evolution law [52].

Eq.(71) implies that all operators are ultimately locally generated: any operator that commutes with all local operators outside $B$ can be generated by products, linear combinations, and limits, of local operators in $B$. In this sense, it is then clear that the notion of non local operator is relative to a region $R$ with some particular topology. An operator can be non local in a ring, but local in a ball that is sufficiently big to contain the ring. That Wilson and 't Hooft loops in non-abelian gauge theories are ultimately generated by local operators was proven by

---

[16]If $a$ is a non local operator in region $R$, then $\mathcal{O} a$, with $\mathcal{O}$ being a local operator in $R$, is still non locally generated in $R$. Therefore non local operators generate classes through a quotient by the additive algebra [13].

[17]Notice this is not a violation of causality since we are essentially discussing features of the additive algebra alone, which is causal. We further comment on this below.

explicit construction in the lattice [12], appendix B2.[18] For the WL this construction turns out to involve both "magnetic" plaquette operators and local gauge invariant "electric" operators on the surface. As was described previously, for the models in the present paper a more direct reason can be given. The Wilson loops are generated by fluxes of the magnetic field, so they are evidently locally generated in a ball. The existence of the dual 't Hooft loops in the ball then follows from von Neumann's double commutant theorem as above. In fact, using the split property, we can restrict attention to a full algebra of operators (a type I factor) restricted to local operators in a regularized ball, instead of considering the full space-time.

It is quite remarkable that the additive net of algebras contains in itself all the physical manifestations of the generalized symmetries. These features appear as textures of the additive algebra, and these textures are a physical and local phenomena. In particular, there is no need to add by hand "non-local external probes". The full set of non local or HDV operators appear when taking commutants inside the additive algebras themselves, and they are dynamical operators that belong to the theory.

One of the main results in [12, 13] is that HDV operators provide a unified definition of what an order/disorder parameter is in a QFT. This was backed up first by explicit examples. But strong support comes by showing that HDV operators are the only types of operators which can exhibit a "generalized volume law" type behavior, where this terminology should be understood in a generalized sense, see [12]. For example, HDV line operators are the only line operators which can exhibit area law. In the reverse direction, if one finds an operator that can show a "generalized volume law", this operator should violate Haag duality in the appropriate region. For example, if one finds a line operator exhibiting area law, then this operator cannot be generated locally in the loop where it is defined.

## 3.2 Genuine lines and topological surfaces as HDV operators

Now we describe the standard way in which generalized symmetries are usually described, connecting it with the previous discussion. In the seminal reference [14], see also [57], a particular classification of the operators/defects appearing in QFT's with generalized symmetries was put forward. This classification is mostly studied in the Euclidean formulation of the theory. The main roles are played by "symmetry generators" and "genuine charged operators", such as genuine lines for one-form symmetries. These genuine charged operators are the order parameters of the generalized symmetry in this formulation.

Albeit sometimes dubbed symmetry topological "operators", it is more proper to say that these are endomorphisms of the operator algebra. Indeed, they are typically defined by their action on the operators. For example, for the $U(1)$ electric 1-form symmmetry in 4$d$ Maxwell theory, the exponentiated electric flux over closed two dimensional surfaces is such a symmetry endomorphism $F_g \equiv e^{i g \Phi_E}$. It acts on Wilson lines $W_q \equiv e^{i \oint A dx} = e^{i q \Phi_B}$ that link with such surface as

$$F_g(W_q) = e^{i g q} W_q. \tag{72}$$

The reason $F_g$ is not a proper operator in the real time Lorentzian theory is simple. For a closed surface the operator we get is the identity due to the Gauss law. But while this is more properly an endomorphism, it has an avatar on the operator algebra of the theory. By cutting the surface in two halves, we get an actual operator, call it $T_g$, in one half and its inverse in the other half. The meaning of the endomorphism in the Lorentzian theory becomes

$$F_g(W_q) = T_g W_q T_g^{-1} = e^{i g q} W_q, \tag{73}$$

---

[18]Known forms of the non abelian Stokes theorem, see [53–56], have this same spirit, but they are inconclusive from the present perspective because they express the Wilson loop in terms of non gauge invariant quantities in the surface bounded by the loop.

where $T_g$ are sometimes called topological surface operators. Of course, once we cut the flux in two halves, local ambiguities appear in the definition of the operator at its one-dimensional boundary.[19] But these ambiguities by the action of local operators do not affect the previous transformation law since they commute with the Wilson loop. In fact, since any representative is as good as any other, it is more proper to talk about the classes that arise by the quotient of the non local operators over such actions of local operators in a certain topologically non trivial region.

We conclude that in real time, the characterization of operators put forward in [14] concerns open fluxes of generalized currents (or more generally topological surface operators) and genuine charged operators.[20] We note, however, that the role of WL and TL in (73) can be inverted. We can as well say that the TL are the charged ones under the action of the WL. They play a dual symmetric role and indeed they can be seen as quantum complementary variables in a precise sense [46]. These operators furnish the $a$'s and the $b$'s in the preceding discussion.[21]

As such, this nomenclature would differ very little from our approach. The problem comes when this particular selection of what is a "genuine line operator" and what is a "topological surface operator" is promoted to have intrinsic physical meaning. For example, what is called the free compact Maxwell field it is said to have genuine Wilson loops $W_q$ in $R$ and genuine 't Hooft loops $T_g$ wrapping the ring $R'$. These are labeled as $q = q_0 n$ and $g = g_0 m$, with $n, m$ being integers and $q_0 = 2\pi/g_0$, saturating the Dirac quantization condition. They obviously commute with the local algebra outside, and they cannot be generated by local gauge invariant operations in $R$ or $R'$ respectively. They are HDV operators in the sense described above. But we also have the topological surface operators, namely the exponential of the electric and magnetic fluxes over open surfaces with boundary $R$ and $R'$. These topological surfaces are labeled by two angles $q \in [0, q_0)$ and $g \in [0, g_0)$ and they precisely do not commute with the previous line operators. These flux operators are also HDV operators, non local in that precise sense. In fact, this model is the ordinary Maxwell field that for $d = 4$ has a group of HDV operators $\mathbb{R} \times \mathbb{R}$ given by arbitrary electric and magnetic charges. In this example, it is quite evident the arbitrariness of the choice of what is called line and topological surface operators. All non local operators are line operators in the sense that they commute with local operators outside the ring in which they are defined, and both are topological surface operators in the sense that they are locally constructible inside a ball, but not a ring. But the same can be said in more complicated examples such as a $SU(2)$ gauge theory.

More than the terminology of line operators and topological surface operators (that usually enter the formulation as endomorphisms and not as operators), the core of the problem seems to be the (artificial) requirement that non local operators corresponding to complementary regions should commute.[22] This is why they are thought of as line operators, the underlying idea is that they safely commute to each other at spatial distance. In fact, in any theory with HDV classes, we can take the step of including in the additive algebras $\mathcal{A}_{\text{add}}(R)$ some non local operators (not all), and take care that the ones we add for complementary regions commute with each other. This way we may arrive at nets of algebras that satisfy Haag duality. These nets are local in the second sense mentioned above. An example is the net of the Maxwell field where we take WL and TL with quantized charges satisfying the Dirac quantization condition. We have called Haag Dirac (HD) nets to this type of choices since the two notions, that of Haag

---

[19] The same is true about the Wilson loop.

[20] For conventional 0-form symmetries, these open fluxes are the local twists we have been discussing previously.

[21] The perspective of this paper is that one can understand the generalized symmetries directly in the local physics of flat space. In an approach called "Symmetry Topological Field Theories" [58–61], extra dimensions are added in order to characterize the symmetries. It would be interesting to understand the connection to such approach.

[22] We see here the difference between the two senses of locality mentioned above in QFT's with generalized symmetries.

duality and Dirac quantization turn out to coincide. Although there is no problem in doing that, we remark this is a purely academic game without any physical consequence. Indeed, we have a) There are always many different possible choices of HD nets. b) The theory is exactly the same for any of these choices, it is not possible to distinguish them physically, since all the nets have the same algebras of local operators with the same expectation values, and all operators ultimately belong to the additive algebra. c) A Haag Dirac net does not satisfy additivity. So, even from a purely mathematical point of view, the additive net, the non local operators, and any other choice of net (such as a different HD net) can be reconstructed from any one of them.

Of course, the choice of a HD net is important if one is going to introduce dynamical charges that destroy the non-additivity of the non local operators.[23] Causality implies that the only way to do it is by breaking non local operators that commute to each other. But the introduction of dynamical charges would produce a different model, and would precisely destroy all generalized symmetries or HDV sectors. The existence of generalized symmetries is the same thing as the existence of different HD nets for the same theory. This is why insisting in a choice of a HD net precisely obscures the nature of the phenomenon.

In this sense, we think it is misleading to say there are two pure gauge theories with gauge groups $SU(2)$ and $SO(3)$ because one of them contains the fundamental WL and the other the TL.[24] They are the same $SU(2)$ theory, containing both the WL and the TL. If there is one non local operator the existence of the dual one cannot be avoided. The naming then refers in fact not to theories but to HD nets. This can be harmless only if no physical consequences are drawn upon it.

This way we learn various things that are obscured in the generalized symmetry literature. The first is that topological surface operators are, from a precise physical perspective, not so much "surface operators" after all. They are naturally associated to their boundary subregions since they commute with the additive algebra outside. The second is that genuine lines are not so much "line operators" after all. To construct them using gauge invariant local operators we need surfaces that go beyond the boundary subregions over which they were defined at first. The third is that the non-commutativity between dual HDV operators is not only not a problem, but a mathematical necessity. It is what allows the topological symmetry endomorphism to do its job on the charged operators. These non-commutative structures directly appear from the additive algebra, which is the intrinsic local algebra, just by analyzing its commutants.

Again we find the remarkable result that the additive algebra itself contains all the information about the order parameters and the symmetry generators. In fact, it becomes clear all HDV operators should be considered as order parameters. This solves a problem already described in the original reference [14], where it was acknowledged that $SU(N)$ gauge theory contains further order parameters than the usual Wilson lines. Basically, these are the non-abelian electric fluxes, the generators of the 1-form symmetry of the theory. Such reference remarked that a more unified theory of order parameters would be desirable. The present approach precisely provides such a unified framework. The solution is that Wilson lines and electric fluxes are both HDV operators for rings in gauge theories, and all of them are order parameters of the theory alike. Moreover, as mentioned before, the only operators that can

---

[23]The choice of a HD net is not necessary if the charges are external probes and are not dynamical.

[24]This terminology is unrelated to the natural terminology in lattice formulation of gauge theories. Note that a lattice $SO(3)$ theory that does not have any non local operator. Equivalently it does not have generalized symmetries and does not display phases in which lines operators have an area law. At least from the lattice perspective this theory should correspond to an $SU(2)$ theory with charges in the fundamental representation. In fact, an important problem in this context is to ascertain whether $SO(3)$ lattice gauge theory has a confining order parameter in the continuum limit [62]. Although charged particles will necessarily appear at some energy scale, screening the area law of the WL, these charges may appear at the scale of the lattice spacing, remaining hidden from the continuum physics. At any rate, the difference between pure $SO(3)$ and $SU(2)$ lattice gauge theories is physical in the lattice.

show "area vs. perimeter laws" are the HDV operators [12].

Finally, we remark that these features appear in Minkowski space, we do not need to go to manifolds with non-trivial topology. However, all the same physics also appears in manifolds with non-trivial topology when looking inside subregions given by contractible balls.

We end this section with some remarks on three issues that may wrongly induce to think a definition of a QFT requires the choice of a Haag-Dirac net.

### 3.2.1 Ambiguities in the Euclidean time ordering?

The translation of quantities computed with the Euclidean path integral to the operator language involves a time ordering. This is required to account for the non commutativity of operators in the quantum theory as opposed to the commutativity of insertions in the path integral. As it is well known, for local operators $\phi(x)$, the Euclidean correlator for both field orderings

$$\langle \phi(x)\phi(y)\rangle = \langle \phi(y)\phi(x)\rangle\,,\qquad(74)$$

computes the following expectation value

$$\langle 0|\phi(\vec{x})e^{-H(t-t')}\phi(\vec{y})|0\rangle\,,\qquad(75)$$

where we assumed $t > t'$. For this prescription to make sense, operators inserted at $t = 0$ and different positions must commute with each other. This is of course the case in QFT.

In theories with generalized symmetries, this Euclidean prescription may appear paradoxical. Imagine we are to compute a correlator between a Wilson loop $W_q$ and a 't Hooft loop $T_g$. We can take both of them at time $t = 0$ and linked to each other. From the Euclidean path integral prescription one may conclude that

$$\langle W_q T_g\rangle = \langle T_g W_q\rangle\,.\qquad(76)$$

But this is of course incorrect since the two terms in general differ by a phase factor. But what is the Euclidean path integral computing here, the left-hand side or the right-hand side?

To sidestep this problem one may simply take the view that only one of the two operators can be inserted in the path integral as a line operator. This may lead to the idea that a HD net is necessary for the theory to have an Euclidean description, because non local operators in a HD net are precisely taken to commute with each other. And if we only allow those we do not run into the previous problem.[25]

But again this is not correct. The problem is simply that the path integral may not understand the meaning of either of $W_q$ or $T_g$, or both, if we are not more precise. To compute the path integral we have to express the integrand in terms of the integration variables. In the usual description of the Maxwell field we path integrate over $A$, and the meaning of the WL is clear enough as an exponential of the circulation of $A$. The TL has then to be written as a flux of the electric field. To avoid coincidence points of operators we can move the TL flux infinitesimally to the future or past, and this will compute either of the two possible orderings. But this was only a prescription that gives meaning to the required path integral computation. An indeed to solve the same problem we could have as well write the WL in terms of the magnetic flux. This way we can get different prescriptions to give meaning to the Euclidean path integral in QFT's with generalized symmetries. Of course, in the Lorentzian QFT this issue does not appear, and there are no ambiguities in the computation of expectation values of products of WL and TL.

---

[25]Further elaboration around this issue was described in [63].

But here we may again be assaulted with a doubt. Why is the position of the flux in time important if the flux is conserved? The answer is that the path integral makes a specific calculation whose interpretation in terms of operators may differ even if it involves the same operator, and depends specifically on how this operator is written in terms of fields. The same happens with the ordinary time ordering in real time. This time ordering is not a map of operators into operators, because to understand what is the result of the time ordering we need to write a given operator in a concrete way in terms of field operators at given times. For example, $\phi(x,t)$ is the same operator if we express it at a different time using the equations of motion. However, we cannot expect that the result of the time ordering with another operator would not depend on these two ways of writing $\phi(x,t)$.

Summarizing, the main physical lessons here is that we can use non commuting non local operators in the path integral without trouble at the cost of remembering that they can be written in terms of local operators, and that each particular prescription in the expression of the non local operators in terms of local fields inserted in the Euclidean path integral computation has its own physical meaning.

### 3.2.2 The (misleading) lessons from orbifolds and 2$d$

Suppose we have a theory $\mathcal{F}$ without Haag duality violations, and with an unbroken global symmetry group $G$. Take the orbifold theory $\mathcal{O} = \mathcal{F}/G$. The orbifold has non local sectors corresponding to regions with the topology of two disjoint balls and with the complementary topology [64]. The non local operators are charge-anticharge operator in the two balls, and twists operators with boundary outside the two balls. Again we can choose HD nets, for example, taking all charge-anticharge operators for the algebra of any two balls, but not taking the twist operator for the algebra of the complement, or the opposite choice.

The point is that for this particular type of orbifold sectors something special happens. We can think in another theory, namely the theory $\mathcal{F}$, containing the charged operators. This is an extension of $\mathcal{O}$ which respects the dynamics and expectation values of $\mathcal{O}$ for the neutral operators, while changing the notion of additive algebra. The charged operators are obtained from the orbifold just by taking a charge-anticharge pair and sending one of then to infinity. The extension $\mathcal{F}$ does not have HDV sectors. Both non local operators, charge-anticharge pairs and twists, live in $\mathcal{F}$. However, while the charge anticharge operator is still an operator (now locally generated) in the algebra of the two balls, the twist cannot be thought as an operator in the complement because it does not commute with charged operators in a single ball. This extension $\mathcal{F}$ is complete in the sense that both Haag duality and additivity are satisfied for any given region, no matter its topology. This notion of completeness coincides with the notion of completeness of the spectrum of charges in gauge theories and the absence of generalized symmetries [13].

One may be tempted to think there was a preferred choice of HD net for the original orbifold $\mathcal{O}$, namely the one in which we chose the charge-anticharge operator for the algebra of the two balls. This choice is the one that allows the non local charge-anticharge operator to be broken into the independent charges, leading to the new theory $\mathcal{F}$. In particular such a completion could not be attained by breaking the twists for $d > 2$. This is the context of the DHR theorem[26] that allows to repair these particular form of Haag duality violations by enlarging the algebra with charged operators. Given the orbifold, this completion is unique for $d > 2$. Notice also that what the DHR shows is that all the information about $\mathcal{F}$ was already in $\mathcal{O}$.

---

[26]A brief description of the DHR theorem will appear in the discussion section below.

For CFT's in $d = 2$, the regions where the non local operators appear consist in two intervals, and the complement has the same topology. It is usually stated that "the orbifold"[27] has a new field operator, namely the twist. This is true if the twist over an interval is broken into its two end points. This indicates that different complete theories (in the sense described above) may be formed by choosing particular HD nets in $d = 2$, and promoting non local operators to local field operators.[28] These different theories coming from different HD net choices are not compatible with each other because the types of broken non local operators that would act as local field operators cannot commute with each other. In this scenario, the fact that these are different theories can be seen already in flat space at a local level: the additive algebra is different and both additivity and Haag duality are satisfied at the same time for any region. Still, notice that the full information about both completions is already in the theory $\mathcal{O}$. Equivalently, from any choice of net, including the net defined by $\mathcal{O}$, we can go to any other choice of net without any further input.

These $2d$ examples may induce one to think that all theories may have completions that are intrinsic to itself and that the importance of HD nets resides in the possible completions of this kind. However, no such an intrinsic "completion for free", that does not change the dynamics of the theory, seems possible for sectors corresponding to other topologies. It is not possible to break open a Wilson loop and place it in the time direction, so as to imitate a charged particle. Such a time-like Wilson line does not have any dynamics that can be dictated by, or would be in agreement with, the one of the original theory. Nor it seems possible to construct Wilson line operators to break the non local WL without introducing actual charged operators that change the theory. Notice also that, while in $2d$ different HD nets where the non local operators are broken to local fields have different additive algebras, the same does not apply to other types of HD nets for generalized symmetries associated with different topologies. For example, a HD net for the Maxwell field does restore Haag duality but the theory still violates additivity, and the additive algebra after such a choice remains the same. We will further comment on this below.

### 3.2.3 Compact manifolds

As we have explained above, the local manifestations of generalized symmetries are generally neglected because they remain hidden by the usual description. As a consequence of this, it is generally asserted that to understand the subtle consequences of generalized symmetries it is necessary to put the theory in topologically non trivial manifolds $M$. Interesting observables may be constructed in this way. But as far as we do not consider gravity theories, putting the theory in a different manifold, even with a Lagrangian formulation, is not an automatic and uniquely defined process. As far as the theory in these manifolds can be understood as the "same theory" originally defined in Minkowski space, these observables should have a Minkowski space understanding, or new data, beyond the one defining the original theory, must constitute an input in the definition of the theory in $M$.

For theories with generalized symmetries, where the phenomenon in flat space is clear enough in terms of HDV classes, we have a combination of these two options in the process of putting the theory in topologically non trivial manifolds. On one hand, the local structure of the algebras must be the same as in flat space, and this can be detected by looking at the HDV sectors inside a ball. In this sense, if the theory is an $SU(2)$ pure gauge theory, this can be distinguished from a theory without sectors by the existence of both the TL and the WL for local algebras. On the other hand, for theories with HDV sectors, the global structure generally

---

[27]Strictly speaking, the orbifold of a theory $\mathcal{F}$ in any dimension is the theory $\mathcal{O} = \mathcal{F}/G$ which does not include neither the charged operators nor putative local twists fields in the case of $2d$. In the literature of $2d$-CFT's the name "orbifold theory" corresponds to adding the local twists fields to the theory $\mathcal{O}$.

[28] Completeness for $d = 2$ CFT is related to modular invariance, see [65–67].

needs to be specified further by arbitrary choices in $M$ (boundary conditions, superselection sectors). To make explicit the ideas described in this section, a simple example of these choices for non trivial manifolds is worked out in section 3.4 below.

These choices are in fact quite similar to the choices we have for the theory in Minkowski space, if we restrict our attention to a subregion $R$. There are in general several different algebras we can assign to $R$, either containing or not some subgroup of non local operators. As we expect to have a type I algebra for a theory in a compact manifold, the parallel is even greater if instead of considering algebras assigned to regions of the space we consider algebras assigned to local type I factors. Using the split property, these can be localized to be larger than the algebra of $R$ and smaller than the algebra of a $\tilde{R}$ slightly greater than $R$. These factors again may be defined to represent any of the algebra choices [11]. For type I factors there is the additional ingredient that the endomorphisms on the algebra effected by dual non local operators in $R'$ are now inner, that is, they are implemented by operators in the algebra itself. Then, we can have both the algebra of non local operators of $R$ and the ones of $R'$ as part of the type I factor.[29] On top of that we can also get type I algebras with center just by eliminating (with a conditional expectation) the global non local operators contained in the original type I factor.

In putting the theory in a non trivial manifold $M$ we again have a type I factor algebra, and we will have, along the possible non local operators $a$ corresponding to the topology of $M$, the dual ones $b$, corresponding to the endomorphisms of this non local algebra. These dual operators commute with the the additive algebra, and with the stress tensor. Then, these dual operators $b$ act as symmetries of the global Hamiltonian, without any action on the local algebras, and the only non trivial action is in the $a$ operators. In consequence, the Hamiltonian is a function of the local operators and the $b's$, which act as conserved charges, but do not depend on the $a's$.

While the local algebras for subregions with non trivial topology in a ball inside $M$ cannot be made unique without changing the theory, the global choices (possible subalgebras of the non local operators and their duals) represent physically different models in $M$. In the standard literature it is insisted that the theory should define a model for any $M$. This of course can be set up, but the physical relevance is again disputable. One can prepare different systems, with different global structure, for different $M$, and there is no connection between these choices as we change $M$. The standard recipe is that one should choose the models for different $M$ according to a single HD net for the original theory. For example, for an $SU(2)$ theory, for any $M$ with a non contractible loop, one can take either the prescription with the global WL or without the global WL, and this is in correspondence with the two HD nets. Again, from the point of view of the original theory, and from a physical point of view, there is nothing pointing to these choices, nor to any prescribed compatibility for the choices for different $M$.[30]

The logic behind this idea that one should choose a single HD net for all manifolds $M$ may come from gravity models, where space is dynamical, and in principle the physics on any manifold should be automatically determined. We are not discussing gravity here. But even in this case, it is not clear why the theory on each manifold should be dictated by a HD net, and not any other choice, or even a quantum superposition of choices. This is more so since the local structure is still enjoying all its arbitrariness. Perhaps this impossibility of eliminating the HD

---

[29]This is not possible for the usual algebras attached to subregions, which are type III. The dual operators in $R'$ commute with the additive algebra in $R$. We can push this operator towards $R$ trying to put it inside the algebra of $R$, but the only way it still commutes with the local algebra is to place it at the boundary of $R$, and in that case the operator becomes singular.

[30]Requiring a QFT to be defined automatically for all manifolds without external input in part also originates in Segal's definition of 2d CFT [68]. This definition prescribes a unique partition function for any manifold, and because of that assumes modular invariance. However, the existence of different HD nets is associated to the failure of modular invariance. See references in footnote 28.

net arbitrariness is related to the usual understanding that ultimately generalized symmetries must be actually absent in gravity theories.

Another motivation may come from connecting the idea of generalized symmetries with the description of topological models. Indeed, large enough manifolds can be used to study the infrared physics of a theory with local degrees of freedom, and this limit may display topological properties. In this case, what happens is that the dynamics of the theory itself determines an effective IR HD net. For example, an $SU(2)$ theory may have confinement, such that at the IR the expectation value of the WL is 0 while the one of the TL is 1 (after suitable smearing). The opposite choice holds for spontaneous symmetry breaking. This saturation of expectation values, either to 1 or 0, can only happen for HD nets, precisely because of the non trivial commutation relations of non local operators, and the quantum uncertainty relations they satisfy. This phenomenon of IR saturation can of course be studied in flat space as well. In this case there is a dynamical choice in the IR. But to study the theory outside the IR purely topological limit, we have to understand that both dual operators exist on equal footing, albeit with different statistics. In the topological limit, the operator with expectation value 1 can be effectively assimilated to a number (the "symmetry generator" that respects the vacuum), while the one with expectation value 0 will lead to degeneracies of the vacuum in a non trivial manifolds. Then, we have rather a structure of superselection sectors corresponding to the different vacua. In the case of the IR limit of confinement, for example, the usual terminology of $SU(2)$ versus $SO(3)$ implies naming the theory by the choice of considering all superselection sectors (and the WL that change between them) or just considering one of them, while all sectors really exist in both cases. This clearly exemplifies that artificially restricting the operator content makes little sense.

An interesting question revealed by the present discussion is the following. Given a flat space QFT, what are the ambiguities in putting this "same theory" in compact manifolds? We argued that the presence of HDV sectors always gives place to such ambiguities. The question is if these are all the possible ones, or if there are more. Another related issue is that subregions of flat space do not have as rich topologies as manifolds of the same dimension. This may suggest other properties may be revealed by non trivial manifolds. These, however, for logical rather than physical reasons, must have another manifestation in flat space.

## 3.3 Implications on recent works on the ABJ anomaly

We now describe how the previous features affect recent interpretations of the ABJ anomaly. The first paper that noticed there was an intersting interplay between the chiral symmetry and the TL was Ref. [17]. However, there it was argued that in QED, Adler's $U(1)$ modified chiral symmetry is not really a global symmetry. The reason given is that this symmetry mixes the 't Hooft loop with a topological surface operator, namely the magnetic flux. The definition they provide of global symmetry is the standard one, and the same as in the algebraic literature, namely as an automorphism of the local algebras. The problem is the notion of local algebra. In the cited paper this means the additive algebra plus the 't Hooft loops, but not the WL. This is making a particular choice of HD net and attaching a physical meaning to it, in this case to decide if some transformations are symmetries of the theory or not.

However, a definition of global symmetry that is physical (intrinsic) must start by automorphisms of the additive algebra alone. An indeed, this is enough for all the discussion to go through. The reason is as follows. Since topologically trivial regions have unique algebras (they satisfy Haag duality), it is a minimalistic requirement that a global symmetry is an automorphism of the algebras associated with topologically trivial regions such as balls. But this minimalistic requirement immediately applies to the whole additive net since it is constructed

from the algebras associated with balls. Suppose then that we indeed have

$$U(g)\,\mathcal{A}_{\text{add}}(R)\,U(g)^{-1} = \mathcal{A}_{\text{add}}(R)\,, \qquad g \in G\,, \tag{77}$$

where this relation is to be understood as a mapping between algebras. Of course, it does not say all elements of $\mathcal{A}_{\text{add}}(R)$ are invariant under the symmetry group. It just says that the action of the symmetry leaves the algebra in itself. Equivalently, if $a \in \mathcal{A}_{\text{add}}(R)$ then $U(g)\,a\,U(g)^{-1} \in \mathcal{A}_{\text{add}}(R)$. Now we recall that conjugation with a unitary $U(g)$ carries commutant algebras into commutant algebras, namely, given (77) we have that

$$U(g)\,\mathcal{A}_{\text{add}}(R)'\,U(g)^{-1} = \mathcal{A}_{\text{add}}(R)'\,, \qquad g \in G\,. \tag{78}$$

But the commutant of the additive algebra in some region is the maximal algebra in the complement, containing all HDV operators.[31] We conclude that if we have an automorphism of the local algebras associated with balls, we have an automorphism for the maximal algebras associated with any region $R$

$$U(g)\,\mathcal{A}_{\text{max}}(R)\,U(g)^{-1} = \mathcal{A}_{\text{max}}(R)\,. \tag{79}$$

This implies that the symmetry cannot convert non local operators into local ones or viceversa. It has to transform the non local operators classes into themselves. Most symmetries do not transform these classes but some do. Since genuine lines and topological surfaces are HDV operators alike, symmetries of this type are expected to transform between those. Notice that the fact that some symmetries can transform genuine lines to topological surfaces is only possible if such distinction is not physically meaningful. In such scenarios, those operators belong to the same generalized multiplet. Equivalently, the notion of HDV classes is the right notion to characterize and classify possible mixings between different generalized symmetries.

For the ABJ-anomaly, as computed above, the commutant of the additive algebra of the ring is the additive algebra in the complementary ring plus the TL $T_g$, with $g = g_0\,n$ and $n$ integer, and the Wilson loops (topological surfaces) $W_q = e^{iq\Phi_B}$, with $\Phi_B$ the magnetic flux over any surface with boundary in the complementary ring, and with $q \in [0, 2\pi/g_0)$. Automorphisms of the additive algebra can be constructed, that leave the additive algebras invariant, but mix the 't Hooft loops with the Wilson loops, as derived in section (2). As the symmetry changes the non local classes it does not respect HD net choices, such as taking only the TL as non local operators for a region. As we have already explained, HD choices are not intrinsic or physical, and a physical symmetry can change them. This type of mixing is the one generated by the modified chiral symmetry. This is a $U(1)$ transformation with no Noether current due to the theorem in [11].

In another view of the subject, in Refs. [18,19], it has been argued that the correct statement is not that of Adler, namely that the ABJ anomaly is codifying an abelian $U(1)$ global symmetry, but that its origin lies in the existence of a certain non-invertible symmetry generated by a modified set of symmetry generators. In the original construction the new symmetry operators were labeled by rational numbers, while extensions to continuous compact labels have been described in [20,21]. The construction follows the line of an analogous one for the case of duality symmetry and related scenarios [22–28]. The motivation for these new developments is that Adler's construction of the conserved gauge invariant charge $Q$ is only valid when the charge is supported by the infinite equal time Cauchy slice, and only when we insert local operators, such as the pion field or $F_{\mu\nu}$ in axion electrodynamics. While it is recognized that in flat space with local operator insertions the symmetry becomes a $U(1)$, the non trivial transformation of the TL is again considered a problem. The unconventional mixing between genuine lines (here the TL) and topological surfaces (the WL with non integer charges) is argued to be related to the proposed non-invertible nature of the symmetry.

---

[31]Equivalently, it contains all genuine lines and topological surfaces.

However, as we have explained, the symmetry does what it does to the TL exactly because it transforms the local fields in the way it does. The TL and the WL are just operators constructed with the local fields.[32] The statement that a symmetry can act in some way in local operators but in a fundamentally different way in non-local operators is inconsistent in QFT. This first principle observation becomes completely explicit in the case of pion electrodynamics, where the 't Hooft loop reads

$$T_g = e^{i\,g\,\Phi_G}\,, \qquad \Phi_G \equiv -\int_\Sigma ds_i\, p_i^A = -\int_\Sigma ds_i \left(\frac{1}{e^2}\,E_i + \frac{1}{\mu}\,\pi_0\,B_i\right). \tag{80}$$

Inserting a 't Hooft loop means nothing more than inserting a bunch of electric, magnetic and pion fields. The non-trivial transformation of the 't Hooft loop follows entirely from that of the local pion field, which in turn is a $U(1)$, as acknowledged by [18, 19]. It is also transparent that the fact that the transformation mixes the 't Hooft loop with the topological surface has nothing to do with the non-invertibility of the symmetry, just with the fact that the 't Hooft loop, in order to be topological, needs to be dressed with the local chiral field. It is transparent that the classification between genuine lines and topological surfaces just causes confusion here, as these models precisely show, connecting them by a symmetry transformation.

These papers then discard to a certain extent the discussion of what is happening for the local algebras and local physics and, according to the standard practice in the subject of generalized symmetries, concentrate to define and understand the symmetry in topologically non trivial manifolds. This takes us away from the realm of the ABJ anomaly and particle physics. For non trivial manifold topologies, the relevant physics inside balls on these manifolds should follow the one we have described in flat space.

As we have mentioned, for the global structure there are additional choices that have to be made to define the model. The global action of the symmetry has to be defined accordingly. We will elaborate more on symmetries that change classes for non trivial manifolds in the next section. Here we want to stress that a very similar problem, with parallel choices and solutions, can be studied inside flat space. We may wonder on how to construct twists for the symmetry that act on a compact region $R$ of the space and do nothing on the complement of a slightly bigger region. In the present case the interest is for example in a region $R$ with the topology of a ring. As was studied in detail in [11], and briefly reviewed in section 2.5, for a topologically non trivial region $R$ there are different choices of algebras and twists we can consider. The algebra can be purely additive, or contain all, or some subgroup of non local classes of operators. Twists for these algebras can be constructed such that they form a $U(1)$ group, and such that they have, at least, the same action as the global symmetry on local operators. If the chosen subalgebra of non local operators is invariant under the symmetry, then twists which make the relevant transformations also on non local operators can also be constructed. These complete twists still form a representation of the original group. We see that no non-invertible action is needed to define the action of the symmetry locally.

## 3.4 Recovering the $U(1)$ symmetry in manifolds with non-trivial topology

In previous sections we have shown that it is possible to recover the $U(1)$ chiral symmetry for contractible subregions of any topology in any manifold. A similar discussion was done in [11] for the electromagnetic duality symmetry of the Maxwell field and subregions of Minkowski space. From our point of view, this is the important statement, since such $U(1)$ symmetries, valid at the local level, are the ones controlling local physics, such as pion decay, in any space-time. But part of the community conventionally defines a global symmetry so that topological

---

[32]This is the case for TL and WL whose defining loops are contractible in the manifold where they are defined. These types of contractible loops are related to the local physics. In the next section, we comment on the non-contractible ones.

symmetry generators exist when the theory is placed in manifolds with non-trivial topology. We now show how to extend the construction there.

For concreteness and simplicity we focus on the example considered recently in [26, 28], namely, the electromagnetic duality transformations of a free Maxwell field in a spacetime $M \times \mathbb{R}$, for the spatial manifold $M = S_2 \times S_1$. This symmetry, as the chiral symmetry, transforms classes, since it rotates the electric and magnetic fluxes of the Maxwell field. Because of this it is considered to become non-invertible as well, see [26, 28] and references therein. We now show how to recover the $U(1)$ duality symmetry in this spacetime. Here we just review the important conceptual features, the details of the computations can be found in appendix (B).

We first analyze the structure of the algebra and then the symmetry transformations. We have then to define the Maxwell theory in this manifold. In flat space, this has a Lagrangian

$$\mathcal{L} = -\frac{1}{4} F_{\mu\nu} F^{\mu\nu}, \tag{81}$$

so it is natural to start with the same Lagrangian. This is symmetric in curved space by rotating $F$ and $F^*$, and we expect this symmetry would be still present. The symmetry interchanges WL and TL and then it cannot have a Noether current. The transition to the Hamiltonian formalism and quantization are straightforward. We get a series of decoupled harmonic oscillators,

$$\left[ q_{lmk}^{(1)}, p_{lmk}^{(1)} \right] = i\,, \qquad \left[ q_{lmk}^{(2)}, p_{lmk}^{(2)} \right] = i\,, \qquad \left[ q_0, p_0 \right] = i\,, \tag{82}$$

in terms of which the Hamiltonian writes

$$H = \frac{1}{2} \int_0^{2\pi} d\chi \int_0^{\pi} d\theta \int_0^{2\pi} d\varphi \sqrt{|g|} g^{ij} \left( E_i E_j + B_i B_j \right) \tag{83}$$

$$= \frac{p_0^2}{2} + \frac{1}{2} \sum_{l=1}^{\infty} \sum_{m=-l}^{l} \sum_{k=-\infty}^{\infty} \left[ (p_{lmk}^{(1)})^2 + (p_{lmk}^{(2)})^2 + \omega_{lk}^2 (q_{lmk}^{(1)})^2 + \omega_{lk}^2 (q_{lmk}^{(2)})^2 \right].$$

The labels $l, m$ correspond to $S^2$ eigenfunctions and $k$ to the momentum in the $S^1$ direction. The coordinates $\theta, \varphi$ are usual spherical coordinate son $S^2$ and $\chi$ is the coordinate along $S^1$. Except for the mode $q_0, p_0$, we see there is a duplication on the modes. This corresponds to the two helicities of the field, related by duality transformations.

The global modes $q_0, p_0$ correspond to constant components of the field

$$q_0 \sim \int_M (A \cdot \hat{\chi})\,, \qquad p_0 \sim \int_\Sigma \star F\,, \tag{84}$$

where $\Sigma$ is the sphere $S^2$ at a position in time and at a point of the $S^1$. Then $q_0$ is proportional to the global generator of Wilson loops along the $S^1$ direction. It is a non local operator for the full space. On the other hand, $p_0$ represents the conserved flux of the electric field over the $S^2$. It can be thought the dual TL variable to $q_0$. As we have mentioned, for type I factor algebras the automorphisms of the algebra (as the ones the TL effect on the WL for complementary regions in flat space) are inner, implemented by operators in the algebra itself. This is why we get the dual operator to the global WL inside the algebra here.

As far as the electric field satisfies $\nabla E = 0$ (or in covariant manner $\delta F = 0$), the flux $p_0 \sim \int_\Sigma \star F$ is constant. Therefore, this operator has to commute with local operators, here represented by the other modes different from $q_0$. Then, this is a general feature, independent of the particular details of the geometry of the manifold, or adding interactions that preserve $\nabla E = 0$. In the same way this flux cannot change in time, what explains the absence of the global WL, or $q_0$, from the Hamiltonian. These are general expected features, as was discussed in section (3.2.3).

In contrast, the total magnetic flux over the $S^2$ vanishes, $\int_\Sigma F = 0$. In the same way, we do not have a version of the TL running along the $S^1$. This implies the algebra in this model is generated by the global WL and the smeared fields

$$\int \sigma_{\mu\nu}(x) F^{\mu\nu}(x), \tag{85}$$

where the smearing two forms $\sigma_{\mu\nu}$ satisfies

$$\int_\Sigma \sigma = 0. \tag{86}$$

Local algebras are simply generated by these smearing functions with support in the region. The reason for this asymmetry for the global modes comes from having quantized the theory using the vector potential $A$, such that $F = dA$ even for the global mode, and this is not necessary for having $dF = 0$. Of course, nothing in the Maxwell theory instructed us to set the magnetic flux over the $S^2$ to zero while letting the electric flux to take any value. This entails a choice, analogous to the choice of HD net. Given the theory in flat space, we could as well have defined our Lagrangian formulation using a dual $\tilde{A}$ such that $*F = d\tilde{A}$. This quantization would have given us a model where the electric flux over the sphere is set to zero and the magnetic one is free. Other choices are also possible.

The question is if there is a definition of Maxwell theory in $S_2 \times S_1$ that respects the electromagnetic duality. In fact, there are two canonical ways. First, we can choose to eliminate both flux operators and the non local WL and TL.[33] This is a restriction to the analogous to the additive algebra in the discussions in the previous sections since this algebra does not contain the non local WL or TL and cannot change fluxes. In this theory, the Hamiltonian writes

$$H_{\text{add}} = \frac{1}{2} \sum_{l=1}^\infty \sum_{m=-l}^l \sum_{k=-\infty}^\infty \left[ (p_{lmk}^{(1)})^2 + (p_{lmk}^{(2)})^2 + \omega_{lk}^2 (q_{lmk}^{(1)})^2 + \omega_{lk}^2 (q_{lmk}^{(2)})^2 \right]. \tag{87}$$

This defines a local QFT in $S_2 \times S_1$. The smearing functions of the fields satisfy[34]

$$\int_\Sigma \sigma = 0, \qquad \int_\Sigma \sigma^* = 0. \tag{88}$$

In this QFT there is a $U(1)$ electromagnetic duality symmetry generated by the following charge

$$Q_{\text{add}} = \sum_{l=1}^\infty \sum_{m=-l}^l \sum_{k=0}^\infty \left( q_{lmk}^{(1)} p_{lmk}^{(2)} - q_{lmk}^{(2)} p_{lmk}^{(1)} \right), \tag{89}$$

where the "add" suffix means it belongs to the additive algebra. As expected, this charge rotates the electric and magnetic fields

$$[Q_{\text{add}}, F_{\mu\nu}(x)] = F_{\mu\nu}^*(x), \qquad [Q_{\text{add}}, F_{\mu\nu}^*(x)] = -F_{\mu\nu}(x), \tag{90}$$

where the constraints (88), compatible with this symmetry, are imposed on the smearing functions.

---

[33]More formally this can be done starting with the theory in (83) and first eliminating $q_0$ from the algebra. This is done by a conditional expectation by acting with exponentials of $p_0$. Then, $p_0$ forms a center of the resulting algebra. We can then set $p_0 = 0$.

[34]The constraint that sets the electric and magnetic fluxes to zero could suggest the idea that the local algebras do not contain the Wightman field $F_{\mu\nu}(x)$. This is not so, the only restrictions are in the available smearing functions of this operator valued distribution. This does not imply there are no local algebras. Notice the same can be said for the choice in which only the electric or the magnetic flux are set to zero.

Notice that this charge will transform HDV classes inside the manifold. Equivalently, it will effect the duality transformation on contractible WL and TL inside the manifold since these operators can be constructed from the electric and magnetic fields themselves. In this sence the symmetry still rotates non local classes. Also notice that with this global charge we can construct local twists in the manifold using modular theory, as explained before. To accomplish this we only need the global charge and the fact that it produces an automorphism of the local algebras, as we just showed.

A second possibility is to allow both electric and magnetic fluxes to pierce the $S^2$ surface. Both fluxes commute with the Hamiltonian and the local algebra (the rest of the modes from this perspective). We can also include the global WL and TL. There is no principle that impedes us to do so. We arrive at this parent theory by introducing a new zero mode $\tilde{p}_0$ whose eigenvalues measure the flux of the magnetic field across the spatial sphere. In this "parent" Maxwell theory, the Hamiltonian, electric and magnetic fields write

$$H_{\max} = \frac{p_0^2 + \tilde{p}_0^2}{2} + H_{\mathrm{add}}. \tag{91}$$

One can see this theory as a different representation of electric and magnetic fields, together with their dynamics, in the manifold of interest. It is fully gauge invariant, and it is a less biased representation than the one arising from the gauge potential $A$ or the dual gauge potential $\tilde{A}$.[35] This choice corresponds to the maximal algebra in the manifold.

With this maximal choice, we also have a $U(1)$ electromagnetic duality symmetry. It is generated by the following charge

$$Q_{\max} = Q_{\mathrm{add}} + \left(\tilde{q}_0 p_0 - q_0 \tilde{p}_0\right), \tag{92}$$

where the "max" suffix means it belongs to the maximal algebra. It transforms all non local operators, both contractible and not contractible. This charge leaves the Hamiltonian invariant. It is just a local duality transformation between electric and magnetic fields (90), that now do not satisfy the constraints (88). This recovers the full $U(1)$ symmetry of the Maxwell field in this manifold, interchanging non local classes.

Of course, if we cut the global mode in biased ways we will immediately "break" the $U(1)$ electromagnetic duality symmetry. Besides the ways described above, there are many other ways to do so. For example, one can take the global electric flux operator and not the WL. In this case the algebra has a center. One can also take a Haag Dirac net for the global operators. For example we can choose subalgebras of the global modes such as $\{e^{iq_0 n}\}$ for integer $n$, instead of all operators generated by $q_0$, and similarly for $\tilde{q}_0$. But notice that in this way, first, we only break the $U(1)$ part associated with the global mode. The $U(1)$ associated with the local theory in the manifold, namely the one acting in all other modes, is still intact and cannot be broken. It controls the local physics. This shows that even in biased global choices there is a $U(1)$ symmetry. All these choices can be implemented by an extension of the $U(1)$ acting on the additive algebra to the full Hilbert space. This extension can be taken to be the identity over the global modes. This way there is no need to invoke a non invertible symmetry to see the manifestations of the duality symmetry. Non invertible symmetries can be also constructed, such as a conditional expectation from the chosen global algebra to the additive algebra followed by the usual $U(1)$ symmetry. However, the present example shows we unnecessarily obscure the physics by thinking in this way.

---

[35]Naively one would think that this choice is not possible since we cannot choose to quantize the theory with $A$ and $\tilde{A}$ at the same time. But this reasoning is not correct because a quantum theory does not need to be defined by quantization.

# 4  Thoughts on Noether's theorem and the UV completion of QFT

In this section we further discuss the anomaly in relation to the strong form of Noether's theorem. The conjecture is that any continuous symmetry that does not affect the non local sectors must have a Noether current. In particular, if the QFT is complete at all scales,[36] namely it has no HDV sectors, the strong version should hold. We have seen that this conjecture is not contradicted by the ABJ anomaly. We now briefly explore other, related, cases.

The first example is a current with a non Abelian anomaly, such as the chiral current for a massless quark in some representation of $SU(N)$ gauge theory. This current is anomalous

$$\partial_\mu J^\mu = \frac{1}{16\pi^2} \operatorname{tr} \epsilon^{\mu\nu\rho\sigma} F_{\mu\nu} F_{\rho\sigma} \,. \tag{93}$$

However, in contrast to the Abelian case, there are instantons for non Abelian theories. These introduce dynamical fluctuations in the right hand side of (93) whose integral measure the number of instantons. In consequence the chiral charge is not preserved by the dynamics and the continuous symmetry is explicitly broken [69]. In QCD this is related to the fact that the $\eta'$ meson is not a Goldstone mode. In particular it is not massless in the massless quark limit.

A similar situation happens for QED in $d = 2$ (Schwinger model). Again, one can argue that instantons-like configurations for the $d = 2$ Abelian gauge theory are responsible for the breaking of the chiral symmetry — for a recent discussion see [17]. The model is exactly solvable and equivalent to a free massive scalar field. In terms of the scalar field, one has

$$J_5^\mu = \overline{\psi}\gamma^\mu\gamma^5\psi = \frac{1}{2\pi}\partial^\mu\phi \,, \tag{94}$$

that shows the symmetry is explicitly broken for a massive scalar, and cannot be repaired. In terms of the original QED variables the chiral symmetry has the anomaly

$$\partial_\mu J_5^\mu = \frac{1}{2\pi}\epsilon_{\mu\nu}F^{\mu\nu}. \tag{95}$$

Hence, from (95) we can still try to define a conserved current, by adding a term proportional to $\epsilon^{\mu\nu}A_\nu$ to $J_5^\mu$. In terms of the scalar the two terms in (95) are proportional to

$$\Box\phi = \frac{e^2}{\pi}\phi \,. \tag{96}$$

Therefore, such an improved current would read

$$\tilde{J}_5^\mu = \partial^\mu\phi(x) - \frac{e^2}{\pi}\int d^2y\left[\partial^\mu G_0(x,y)\right]\phi(y), \qquad \Box G_0(x,y) = \delta(x-y). \tag{97}$$

We see we can construct the conserved current at the expense of being non local. One concludes that in these examples there is in fact no ordinary internal continuous symmetry and the absence of a Noether current is no more than expected.

A more challenging example is the following. Take QED in $d = 4$ with a massless fermion of charge $q_1$ and another field, that can be a scalar, of charge $q_2$, such that $q_1/q_2$ is irrational. In this case the chiral charge can be defined as above, but there are no HDV sectors for the Maxwell field. All WL with charge that is combination $n_1 q_1 + n_2 q_2$ with $n_1, n_2$ integer, are locally generated. In consequence there are no non-local TL. The reason is that a non local

---

[36]We are using the word complete with the two different meanings that are ussually assigned to it. We will call complete to a theory without Haag duality violations (with a complete set of charges), while UV complete for a theory that has a well defined UV fix point.

operator in a loop has to commute with local operators outside, and a TL for any charge could not commute with all locally generated WL. So, at first sight we have a symmetry that does not change non local classes but does not have a current. Next we try to understand why the UV completion of this model must have problems.

This theory with a dense set of charges was proposed in [36] as an example of a model where there are non local sectors, here the WL with charge that is not of the form $n_1 q_1 + n_2 q_2$, and the dual ones are absent. But this is impossible [13], because the existence of non local operators is tied to the existence of the dual ones as enforced by von Neumann double commutant theorem. Moreover, the dual non local classes, if Abelian, must be dual groups. Then, in this case what must happen in a complete theory is that the set of charges is in fact the continuous real line. To have a continuum of dynamical charges is in fact a problem that most probably render the theory sick. Let us think in the simpler case of charges for a global symmetry. Such a theory would not have a well defined partition function (or violate the split property). For example, if we put the theory in a compact sphere, the eigenstates must be discrete to have a finite partition function. But a discrete set of eigenstates cannot accommodate a continuous set of charges unless there are infinite degeneracies. In fact, the DHR theorem gives the possible unbroken global symmetries in QFT as compact groups, having discrete, non dense, sets of charges. There are examples of spontaneously broken non compact global symmetries with a continuum of charges, but these are free models [39].

To understand in more physical terms what is going on, we can take the view of effective field theory. Consider energy scales below some given cutoff. To simplify, let us think the electromagnetic field is weakly coupled, and we have two nearly free scalars of non commensurable charges. The charge $m_1 q_1 + m_2 q_2$ is produced by polynomials in the fields, producing an operator with dimension at least $|m| + |n|$. So the operators that break the WL cannot be considered light for all charges, and we certainly do not have a dense set of charges at any scale. In fact, for each fixed scale, we would have many sectors that cannot be considered broken. In this physical scenario, the absence of a Noether current is still explained by the fact that at low energies we have HDV sectors charged under the continuous symmetry. If we want to break these sectors at low energies, we need to introduce a new light field and new degrees of freedom. And to break a dense set of sectors at a given scale we would need too many local degrees of freedom such that the theory would not have a well defined partition function. Summarizing, if we only add two charged field of non commensurable charges, then the arguments in [11] still holds at low energy. If we add a continuous set of charged local fields at low energy, then the theory becomes ill-defined from various points of view.

Finally, instead of considering non commensurable charges, another way to erase the HDV sectors is to include magnetic monopoles saturating the Dirac quantization condition. The group $U(1) \times \mathbb{Z}$ of non local operators then disappears. More precisely the $\mathbb{Z}$ group of 't Hooft loops becomes additively generated because we can create the loops using lines ending on the monopoles, and the $U(1)$ ceases to be topological because $dF \neq 0$ in the presence of monopoles. But this does not challenge Noether's theorem because at the same time, we explicitly break Adler's $U(1)$ chiral symmetry. Since we cannot write $F = dA$, the modified chiral charge will not be conserved. That the symmetry is explicitly broken can also be seen because the chiral transformation would transform, through Witten's effect, a monopole to a dyon that does not exist in the theory.

## 4.1 Two conjectures

These reflections lead us to refine the conjecture that a theory with a continuous global symmetry that does not change the non local classes must have a Noether current. As such, this statement applies to complete theories with an exact symmetry and non local sectors. But in this case, we can also make statements about the IR physics. In particular, if the symmetry

survives at the IR, the current must also be an operator in the IR theory. Then we have a sort of converse of the anomaly matching discussed above

- If an effective model has a continuous symmetry that changes HDV classes (and therefore with no current), then any completion of the model that preserves the symmetry has to preserve the charged HDV sectors.

As the phenomenon of the charged non local classes and the non existence of the current seems to appear in a form independent of a large separation of scales, we can further conjecture that

- If an effective model has a continuous symmetry without current, but no HDV sector is charged under the symmetry, the model does not have a UV completion.

An example of this last class is the generalization of pion electrodynamics to six dimensions. This effective model has an action

$$S = \frac{1}{2} \int d\pi_0 \wedge \star d\pi_0 + \frac{1}{2e^2} \int F \wedge \star F + \frac{1}{\mu} \int \pi_0 F \wedge F \wedge F. \tag{98}$$

This has a local symmetry $\pi_0 \rightarrow \pi_0 + \text{cons}$ that follows from the invariance of the action. As in the $4d$ model, a non gauge invariant current and global charge can be constructed. However, now the TL are three dimensional hypersurface operators, and the symmetry cannot change classes by mixing it with the WL, for the simple reason that WL are ordinary one dimensional loop operators. In fact, one can verify through direct computation that the symmetry only modifies the TL by local operators without changing classes. Therefore, the conjecture states that this model cannot be consistently completed in the UV.

## 5 Discussion

The notion of generalized symmetry is forcing us to rethink traditional problems in the context of QFT and quantum gravity. This work has concentrated on the interplay between generalized symmetries, ABJ anomalies, and Noether's theorem, a topic that has received a great deal of attention recently. Using the notion of HDV sectors, we have clarified why the ABJ anomaly can be formulated in terms of a conventional $U(1)$ global symmetry. The only particularity is that this symmetry transforms the HDV classes. At once, this explains anomaly quantization, anomaly matching, the validity of Goldstone theorem, and the absence of a Noether current generating this symmetry.

In this way we hope to have clarified certain subtle issues concerning the local manifestation of generalized symmetries. There are two issues that are particularly relevant for this problem, as we discussed above. The first is the notion of local algebra of a certain region $R$. An intrinsic physical choice is formed by local operators in the region, namely the additive algebra. One should be careful when assigning strong meaning to other (biased) choices. The second is the fact that any localized operator, including non local ones in certain region $R$, such as Wilson loops, is ultimately generated by local operators. If we are in a situation in which certain property (such as having a particular symmetry) depends on the nature of non local operators, and it cannot be traced back to the nature of local operators, this is a sign of a problem.

There are various venues for future work. Given that theories with ABJ anomalies further support our conjecture concerning the strong version of Noether's theorem, it would be important to have a proper derivation. In this way, this might clarify the fate of the effective theories

mentioned in the last section. More generally, we also expect it will teach us further natural properties a QFT should have to a have a UV completion. Another important venue concerns the fate of non-invertible symmetries, which we discuss now separately.

## 5.1 Towards a generalized reconstruction theorem

We end with a disgression about non-invertible symmetries, the reconstruction theorem in QFT, and a potential generalization thereof. We discuss each topic in turn.

As described above, there has been recently intense activity around the notion of non-invertible symmetry in $d > 2$ dimensions. As their name suggests, non-invertible symmetries are generated by operators that do not satisfy a group theory law. In particular, the "symmetry operations" have no inverse. These are generated by topological operators (or endomorphisms of the local algebras) $T_r$ satisfying more general fusion algebras. On general grounds, these are of the form

$$T_r \, T_{r'} = \sum_{r''} N_{rr'}^{r''} \, T_{r''} \,, \tag{99}$$

where there are more than one $r''$ with non zero fusion coefficient.

Probably counter-intuitively, obtaining theories with these symmetry structures is quite simple. Indeed, earlier examples of non-invertible symmetries in general dimensions were found in the analysis of ordinary non Abelian symmetries in Ref. [12, 64] (for a non mathematically oriented review see [13]). They arise by taking a QFT with a global non Abelian symmetry $G$, and considering the orbifold $QFT/G$. The seed QFT contains, by assumption, invertible topological defects, the twists $\tau_g$ generating the symmetry $G$. In the orbifold, they give rise to non-invertible topological defects $\tau_c$ labeled by the conjugacy classes $c$ of the group $G$. More precisely, since a linear combination of topological operators is a topological operator, by defining

$$\tau_c = \sum_{g \in c} \tau_g \,, \tag{100}$$

we get topological operators, labeled by the conjugacy classes $c$, belonging to $QFT/G$ and satisfying

$$\tau_c \, \tau_{c'} = \sum_{c''} N_{cc'}^{c''} \, \tau_{c''} \,, \tag{101}$$

where $N_{cc'}^{c''}$ are the fusion coefficients of the category of conjugacy classes. We refer to Ref. [12] for further details, including the diagonalization of the algebra and the construction of topological projectors that transparently show the symmetry has become non-invertible. Therefore, $QFT/G$ is a theory with a non invertible symmetry.

Similar examples for 1-form symmetries can be easily constructed. A simple example goes by taking the product of two independent $SU(2)$ gauge theories with a $Z_2$ group of non local operators (say, for simplicity, for $d = 5$) and making the diagonal orbifold. Calling each flavor 1 and 2 respectively, the seed theory had four non local ring-like sectors generated by the Wilson loops and their product $\mathbb{1}, W_1, W_2, W_1 W_2$. The orbifolded theory has only three sectors generated by $\mathbb{1}, W_1 + W_2, W_1 W_2$. In this case, this construction even provides fusion rules that are not related to the category of representations/conjugacy classes of a group. As can be easily verified, the fusion rules of such non local sectors are of the Ising type.[37]

Many of the examples in $d > 2$ that have been considered recently are of this type, namely, they arise by a global quotient $QFT/G$. In these examples, the non invertibility of the symmetry is quite mild since it can be repaired by considering the full QFT rather than the orbifold. We just have an algebra of topological operators (of any codimension) satisfying group-like fusion

---

[37]Notice that one cannot use this construction to provide 0-form non-invertible sectors with fusion rules not related to the fusion of representations/conjugacy classes of a group.

rules and choose particular subalgebras with non trivial fusion rules. This is very different from what happens in $d = 2$, where there are locally non-invertible symmetries, irrespective of global boundary conditions. Still, the interesting question remains as if there are non-invertible 0-form or 1-form symmetries that do not arise from such an orbifold construction. That these exist seems to be the overall present consensus in the community and part of the claim of Refs. [18, 19] is that these exist in theories with ABJ anomalies. We now argue on the contrary.[38]

To this end, it is important to recall the so-called reconstruction theorem in QFT [29–33]. In this language, this theorem precisely states the non-existence of non-invertible 0-form symmetries in $d > 2$ that do not arise from the previous construction, i.e as resulting from a global quotient of the form $QFT/G$. From a more physical standpoint, this theorem states that non-invertible symmetries do not have observable consequences at the local level. The assumptions for deriving this theorem are very mild, basically the notion of Haag duality for ball-shaped regions. This notion just states that every operator is ultimately generated by products of local operators, which is the essence of locality in QFT. The theorem then follows by a subtle interplay between the constraints imposed by relativistic symmetry and Einstein microcausality on one hand, and the category of global (0-form) superselection sectors on the other. The recent results/claims of [18, 19] are therefore in tension with such reconstruction theorem. The same applies to the claims that duality symmetry is non-invertible.

From this perspective, the present article can be interpreted as a clarification of this tension, consistent with the reconstruction theorem. There remains the question as to whether we can have non-invertible one-form symmetries that do not arise by the previous global quotient procedure. We now argue that this is not the case. This should lead to a "generalized reconstruction theorem", to be discussed elsewhere. The idea is as follows. Consider a generic QFT with only 0 and 1-form symmetries, probably some of them being non-invertible. The reconstruction theorem tells that the 0-form part of the symmetry is, at best, non-invertible due to a quotient. We can eliminate the problems arising from such non-local sectors by going to the parent theory, i.e $QFT/G \rightarrow QFT$. This is precisely the reconstruction of a theory with local charges from its neutral sector. But once we have eliminated the 0-form sectors, we can use the construction in [12], section 2.2. There it was shown that a QFT with no 0-form sectors can only have 1-form sectors generated by an abelian symmetry group. This potentially shows that non-invertible 1-form sectors are again ultimately due to an underlying quotient of the form $QFT/G$. One should continue in this way by analyzing higher than 1-form symmetries, potentially leading to a generalized reconstruction theorem.

## Acknowledgments

We wish to thank D. Harlow and H. Ooguri for the suggestion to study anomalous models in order to test our conjecture concerning Noether's theorem. We also wish to thank I. García-Etxebarria for discussions and valuable comments.

**Funding information** This work was partially supported by CONICET, CNEA and Universidad Nacional de Cuyo, Argentina.

---

[38]The case of theories with ABJ anomalies, and why they should be understood as conventional invertible symmetries has been the content of this article. We want to take now a more general path.

# A Pion stress-energy tensor and the quasi-topological WZW coupling

In this appendix, we study the stress-energy tensor of the neutral pion in the presence of the Quasi-Topological WZW coupling introduced as in (A.3). See [70] for a more general discussion regarding Quasi-Topological couplings. To begin, the stress-energy tensor obtained by deriving the action with respect to the metric in $(+,-,-,-)$ signature is given by

$$T^{\mu\nu} = \frac{2}{\sqrt{g}} \frac{\delta S}{\delta g_{\mu\nu}}. \tag{A.1}$$

At this point, it becomes relevant the quasi-topological nature of the interaction term. Considering that the Hodge dual $\tilde{F}$ of the field-strength is conserved

$$\partial_\nu \tilde{F}^{\mu\nu} = \epsilon^{\mu\nu\rho\sigma} \partial_\nu \partial_\rho A_\sigma = 0, \tag{A.2}$$

the action arising from the lagrangian (3) can be conveniently re-written as

$$S = \int d^3x \left[ \frac{1}{2} \partial_\mu \pi_0 \partial^\mu \pi_0 - \frac{1}{4e^2} F_{\mu\nu} F^{\mu\nu} + \frac{\pi_0}{4\mu} \partial_\mu \left( \epsilon_{\mu\nu\rho\sigma} A^\nu F^{\rho\sigma} \right) \right]. \tag{A.3}$$

This evidences that the interaction term in (A.3) does not couple to the metric and therefore has no contribution to (A.1). Therefore we recover only the "free" contributions as in (16). This is

$$T^{\mu\nu} = \left( \partial^\mu \pi_0 \partial^\nu \pi_0 - \frac{g_{\mu\nu}}{2} \partial_\alpha \pi_0 \partial^\alpha \pi_0 \right) + \frac{1}{e^2} \left( F^{\mu\alpha} F_\alpha{}^\nu + \frac{g_{\mu\nu}}{4} F_{\alpha\beta} F^{\alpha\beta} \right). \tag{A.4}$$

On the other hand, the canonical stress tensor is given by

$$\begin{aligned}
\Theta_{\mu\nu} &= \frac{\delta \mathcal{L}}{\delta \partial_\mu \pi_0} \partial^\nu \pi_0 + \frac{\delta \mathcal{L}}{\delta \partial_\mu A_\alpha} \partial^\nu A_\alpha - g_{\mu\nu} \mathcal{L} \\
&= \left( \partial^\mu \pi_0 \partial^\nu \pi_0 - \frac{g_{\mu\nu}}{2} \partial_\alpha \pi_0 \partial^\alpha \pi_0 \right) - \frac{1}{e^2} \left( F^{\mu\alpha} \partial^\nu A_\alpha - \frac{g_{\mu\nu}}{4} F_{\alpha\beta} F^{\alpha\beta} \right) \\
&\quad + \frac{\pi_0}{\mu} \left( \tilde{F}^{\mu\alpha} \partial^\nu A_\alpha - \frac{g_{\mu\nu}}{4} \tilde{F}^{\alpha\beta} F_{\alpha\beta} \right),
\end{aligned} \tag{A.5}$$

which seems to evidence a non-trivial contribution coming from the interaction term in (3). Clearly, $\Theta_{\mu\nu}$ is not symmetric nor gauge invariant but can be improved in the usual manner. To be specific we may use the equations of motion to write that

$$T_{\mu\nu} = \Theta_{\mu\nu} + \partial^\alpha \chi_{\alpha\mu\nu} + \frac{\pi_0}{\mu} \left( \tilde{F}_{\mu\alpha} F_\nu{}^\alpha - \frac{g_{\mu\nu}}{4} \tilde{F}^{\alpha\beta} F_{\alpha\beta} \right), \tag{A.6}$$

where $\chi^{\alpha\mu\nu}$ obeys the usual improvement condition $\chi^{\alpha\mu\nu} = -\chi^{\mu\alpha\nu}$ as it is given by

$$\chi^{\alpha\mu\nu} = G^{\mu\alpha} A^\nu = \left( \frac{1}{e^2} F^{\mu\alpha} - \frac{\pi_0}{\mu} \tilde{F}_{\mu\alpha} \right) A^\nu. \tag{A.7}$$

However, the last term in (A.6) a priori prevents the improvement of (A.5) to (A.4). A closer analysis of this tensor structure evidences that

$$\tilde{F}_{\mu\alpha} F_\nu{}^\alpha = -\frac{1}{4} \epsilon_{\alpha\mu\rho\sigma} \epsilon^\alpha{}_{\nu\lambda\epsilon} F^{\rho\sigma} \tilde{F}_{\lambda\epsilon} = \frac{1}{4} g_{\mu\rho\sigma, \nu\lambda\epsilon} F^{\rho\sigma} \tilde{F}_{\lambda\epsilon} = \frac{g_{\mu\nu}}{2} \tilde{F}^{\alpha\beta} F_{\alpha\beta} - \tilde{F}_{\mu\alpha} F_\nu{}^\alpha, \tag{A.8}$$

which indeed shows that this canonical stress (A.5) is consistent with the one obtained by deriving with respect to the metric (A.4) as

$$\tilde{F}_{\mu\alpha} F_\nu{}^\alpha = \frac{g_{\mu\nu}}{4} \tilde{F}^{\alpha\beta} F_{\alpha\beta} \quad \Rightarrow \quad T_{\mu\nu} = \Theta_{\mu\nu} + \partial^\alpha \chi_{\alpha\mu\nu}. \tag{A.9}$$

The question that remains is if (A.4) is enough to provide the right time evolution. The stress tensor produces the Hamiltonian

$$H = \int d^3x\, T^{00}(x) = \frac{1}{2}\int d^3x\left[p_0^2 - \partial_i\pi_0\partial^i\pi_0 - \frac{1}{e^2}\left(E_iE^i + B_iB^i\right)\right],$$ (A.10)

where we can use (10-13) to obtain that

$$i\Big[H,\pi_0(x)\Big] = p_0(x),$$ (A.11)

$$i\Big[H,A_i(x)\Big] = E_i(x),$$ (A.12)

$$i\Big[H,p_0(x)\Big] = -\partial_i\partial^i\pi_0(x) - \frac{1}{\mu}B_i(x)E^i(x),$$ (A.13)

$$i\Big[H,E_i(x)\Big] = -\epsilon_{ijk}\partial^jB^k(x) - \frac{e^2}{\mu}\Big(B_i(x)p_0(x) + \epsilon_{ijk}E^j(x)\partial^k\pi_0(x)\Big),$$ (A.14)

which is equivalent to the equations of motion. To sum up, the stress-tensor (A.4) is indeed the gauge-invariant symmetric generator of the time evolution of the theory.

# B   Free Maxwell field in $S^2 \times S^1$

In this appendix, for completeness, we describe the details of the calculations of duality transformations in a free Maxwell field on $S_2 \times S_1 \times \mathbb{R}$. The metric of the space-time is taken to be

$$ds^2 = g_{\mu\nu}dx^\mu dx^\nu = -dt^2 + R^2\left(d\theta^2 + \sin^2\theta\, d\varphi^2\right) + L^2d\chi^2,$$ (B.1)

where $t \in \mathbb{R}$ describes the time coordinate, $\theta \in [0,\pi)$ and $\varphi \in [0,2\pi)$ represent the $S^2$ angles with radius $R$, and $\chi \in [0,2\pi)$ is the $S^1$ angle with radius $L$.

It will be useful to find a basis to span scalar and vector fields in the spatial part of the geometry $S_2 \times S_1$. To start with, the eigenfunctions of the spatial Laplacian read

$$\Phi_{lmk}(\theta,\varphi,\chi) = \frac{1}{R}\frac{e^{ik\chi}}{\sqrt{2\pi L}}Y_{lm}(\theta,\varphi),$$ (B.2)

with $l, \in \mathbb{N}_0$, $m, k \in \mathbb{Z}$, and $|m| \le l$. The corresponding eigenvalues are given by the frequencies $\omega_{lk} = \frac{l(l+1)}{R^2} + \frac{k^2}{L^2}$ as

$$\nabla^2\Phi_{lmk}(\theta,\varphi,\chi) = \frac{1}{\sqrt{|g|}}\partial_i\left[\sqrt{|g|}\,g^{ij}\partial_j\Phi_{lmk}(\theta,\varphi,\chi)\right] = -\omega_{lk}\Phi_{lmk}(\theta,\varphi,\chi).$$ (B.3)

The functions (B.2) obey the orthonormality condition given by the scalar product

$$\int_0^{2\pi}d\chi\int_0^\pi d\theta\int_0^{2\pi}d\varphi\,\sqrt{|g|}\Phi^*_{lmk}(\theta,\varphi,\chi)\Phi_{l'm'k'}(\theta,\varphi,\chi) = \delta_{ll'}\delta_{mm'}\delta_{kk'}.$$ (B.4)

Using the scalar modes, we can obtain a basis for vectors on $S_2 \times S_1$, namely

$$\left[\Phi^e_{lmk}(\theta,\varphi,\chi)\right]_i = \frac{e^{ik\chi}}{\sqrt{2\pi L}}\frac{\partial_iY_{lm}(\theta,\varphi)}{\sqrt{l(l+1)}}, \qquad l > 0, \quad -l \le m \le l, \quad k \in \mathbb{Z},$$ (B.5)

$$\left[\Phi^m_{lmk}(\theta,\varphi,\chi)\right]_i = \frac{1}{L}\frac{e^{ik\chi}}{\sqrt{2\pi L}}\frac{E_{ijn}\hat{\chi}^j\partial^nY_{lm}(\theta,\varphi)}{\sqrt{l(l+1)}}, \qquad l > 0, \quad -l \le m \le l, \quad k \in \mathbb{Z},$$ (B.6)

$$\left[\Phi^\chi_{lmk}(\theta,\varphi,\chi)\right]_i = \frac{1}{RL}\frac{e^{ik\chi}}{\sqrt{2\pi L}}Y_{lm}(\theta,\varphi)\,g_{ij}\hat{\chi}^j, \qquad l \ge 0, \quad -l \le m \le l, \quad k \in \mathbb{Z}.$$ (B.7)

In these mode definitions, notice that $\vec{\Phi}^{\chi}_{lmk}$ is parallel to the $S_1$ versor $\hat{\chi}^i = (0, 0, 1)$, and $\vec{\Phi}^{e}_{lmk}$ and $\vec{\Phi}^{m}_{lmk}$ represent the usual "electric" and "magnetic" directions on $S_2$. We have also defined the Levi-Civita tensor as $E_{ijk} = \sqrt{g}\varepsilon_{ijk}$, with $\varepsilon_{\theta\varphi\chi} = 1$, and we note the spatial gradient is defined on the natural coordinates of the manifold $S_2 \times S_1$, i.e $\partial_i = \left(\frac{\partial}{\partial\theta}, \frac{\partial}{\partial\varphi}, \frac{\partial}{\partial\chi}\right)$. Using the previous orthonormality relations associated with the scalar modes, we can verify that the vectors (B.5-B.7) satisfy

$$\int_0^{2\pi} d\chi \int_0^{\pi} d\theta \int_0^{2\pi} d\varphi \sqrt{|g|}g^{ij}\big[\Phi^{s*}_{lmk}(\theta, \varphi, \chi)\big]_i\big[\Phi^{s'}_{l'm'k'}(\theta, \varphi, \chi)\big]_j = \delta_{ll'}\delta_{mm'}\delta_{kk'}\delta_{ss'}, \quad \text{(B.8)}$$

where the indexes run as $s, s' = e, m, \chi$. Further properties can be computed by considering the corresponding action of the covariant derivatives $\nabla_i$. For instance, the curls and divergences take the form

$$E_{ijn}\nabla^j\big[\Phi^e_{lmk}\big]^n = \frac{ik}{L}\big[\Phi^m_{lmk}\big]_i, \qquad\qquad \nabla_i\big[\Phi^e_{lmk}\big]^i = -\frac{\sqrt{l(l+1)}}{R}\Phi_{lmk}, \quad \text{(B.9)}$$

$$E_{ijn}\nabla^j\big[\Phi^m_{lmk}\big]^n = -\frac{ik}{L}\big[\Phi^e_{lmk}\big]_i - \frac{\sqrt{l(l+1)}}{R}\big[\Phi^{\chi}_{lmk}\big]_i, \qquad \nabla_i\big[\Phi^m_{lmk}\big]^i = 0, \quad \text{(B.10)}$$

$$E_{ijn}\nabla^j\big[\Phi^{\chi}_{lmk}\big]^n = -\frac{\sqrt{l(l+1)}}{R}\big[\Phi^m_{lmk}\big]_i, \qquad\qquad \nabla_i\big[\Phi^{\chi}_{lmk}\big]^i = \frac{ik}{L}\Phi_{lmk}. \quad \text{(B.11)}$$

Going back to the case at hand, we can expand the Maxwell field in modes by combining (B.2) and (B.5-B.7). More precisely we can expand the scalar and vector parts of the gauge potential as

$$A_0 = \sum_{lmk}A^0_{lmk}(t)\Phi_{lmk}(\theta, \varphi, \chi), \qquad A_i = \sum_{lmk}\sum_s A^s_{lmk}(t)\big[\Phi^s_{lmk}(\theta, \varphi, \chi)\big]_i. \quad \text{(B.12)}$$

Note that for consistency with definitions (B.5-B.7) we have $l \geq 1$ for $s = e, m$, whereas $l \geq 0$ for $s = \chi$. We will use this abuse of notation throughout the appendix. Also, the fact that $A_\mu(x)$ is a real vector field combined with the fact that $\Phi^{s*}_{lmk} = (-1)^m\Phi^s_{l-m-k}$ and $\Phi^*_{lmk} = (-1)^m\Phi_{l-m-k}$ implies that

$$A^s_{lmk}{}^* = (-1)^m A^s_{l-m-k}, \qquad A^0_{lmk}{}^* = (-1)^m A^0_{l-m-k}. \quad \text{(B.13)}$$

In this formulation, the gauge transformation $A'_\mu = A_\mu + \partial_\mu\alpha$ can be re-written by spanning $\alpha(t, \theta, \varphi, \chi)$ using the scalar mode basis (B.2). We obtain

$$A'_0 = \sum_{lmk}\left(A^0_{lmk} + \dot{\alpha}_{lmk}\right)\Phi_{lmk}, \quad \text{(B.14)}$$

$$A'_i = \sum_{lmk}\left(A^e_{lmk} + \frac{\sqrt{l(l+1)}}{R}\alpha_{lmk}\right)\big[\Phi^e_{lmk}\big]_i + A^m_{lmk}\big[\Phi^m_{lmk}\big]_i + \left(A^{\chi}_{lmk} + \frac{ik}{L}\alpha_{lmk}\right)\big[\Phi^{\chi}_{lmk}\big]_i. \quad \text{(B.15)}$$

From this relation it is transparent we can fix the gauge so as to impose $A^e_{lmk} = 0$ for all $m, k$ and $l \geq 1$. We impose such a gauge in what follows.

From the previous expressions, we can also obtain the electric and magnetic fields, defined respectively as $E_i = -\dot{A}_i + \partial_i A_0$ and $B_i = E_{ijk}\nabla^j A^k$, that describe the gauge invariant phase space. Using (B.9-B.11) and (B.12), we find that

$$E_i = (-1)\sum_{lmk}\left[\dot{A}^m_{lmk}\big[\Phi^m_{lmk}\big]_i - \frac{\sqrt{l(l+1)}}{R}A^0_{lmk}\big[\Phi^e_{lmk}\big]_i + \left(\dot{A}^{\chi}_{lmk} - \frac{ik}{L}A^0_{lmk}\right)\big[\Phi^{\chi}_{lmk}\big]_i\right], \quad \text{(B.16)}$$

$$B_i = (-1)\sum_{lmk}\left[\frac{\sqrt{l(l+1)}}{R}A^{\chi}_{lmk}\big[\Phi^m_{lmk}\big]_i + \frac{ik}{L}A^m_{lmk}(t)\big[\Phi^e_{lmk}\big]_i + \frac{\sqrt{l(l+1)}}{R}A^m_{lmk}\big[\Phi^{\chi}_{lmk}\big]_i\right]. \quad \text{(B.17)}$$

Building on these results, we can re-write the action in the following form

$$S = \frac{1}{4}\int d^4x \sqrt{|g|}g^{\mu\alpha}g^{\nu\beta}F_{\mu\nu}F_{\alpha\beta} = \int dt\,\mathcal{L} = \sum_{l=0}^{\infty}\sum_{m=-l}^{l}\sum_{k=-\infty}^{\infty}\int dt\,\mathcal{L}_{lmk}, \qquad \text{(B.18)}$$

where the Lagrangian modes $\mathcal{L}_{lmk}$ can be computed combining (B.8) and (B.16-B.17). For instance, if $l \geq 1$, we recover

$$\mathcal{L}_{lmk} = \frac{1}{2}\left(\dot{A}_{lmk}^{m}{}^{*}\dot{A}_{lmk}^{m} - \omega_{lk}^2 A_{lmk}^{m}{}^{*}A_{lmk}^{m} + \dot{A}_{lmk}^{\chi}{}^{*}\dot{A}_{lmk}^{\chi} - \left(\omega_{lk}^2 - \frac{k^2}{L^2}\right)A_{lmk}^{\chi}{}^{*}A_{lmk}^{\chi}\right.$$
$$\left. + \omega_{lk}^2 A_{lmk}^{0}{}^{*}A_{lmk}^{0} - \frac{ik}{L}\left(\dot{A}_{lmk}^{\chi}{}^{*}A_{lmk}^{0} - A_{lmk}^{0}{}^{*}\dot{A}_{lmk}^{\chi}\right)\right). \qquad \text{(B.19)}$$

Given this form of the Lagrangian, where all modes are basically decoupled, we can proceed mode by mode. Note that $A_{lmk}^0$ and is conjugate appear as a Lagrange multipliers. The corresponding equations of motion produce the constraints

$$\omega_{lk}^2 A_{lmk}^0 = -\frac{ik}{L}\dot{A}_{lmk}^{\chi}, \qquad \omega_{lk}^2 A_{lmk}^0{}^{*} = \frac{ik}{L}\dot{A}_{lmk}^{\chi}{}^{*}. \qquad \text{(B.20)}$$

This can be seen to be equivalent to the Gauss law $\nabla_i E^i = 0$ by recalling (B.9-B.11) and (B.16). Solving for the constraint in the Lagrangian leads to

$$\mathcal{L}_{lmk} = \frac{1}{2}\left(\dot{A}_{lmk}^{m}{}^{*}\dot{A}_{lmk}^{m} - \omega_{lk}^2 A_{lmk}^{m}{}^{*}A_{lmk}^{m} + \left(1 - \frac{k^2}{\omega_{lk}^2 L^2}\right)\dot{A}_{lmk}^{\chi}{}^{*}\dot{A}_{lmk}^{\chi} - \left(\omega_{lk}^2 - \frac{k^2}{L^2}\right)A_{lmk}^{\chi}{}^{*}A_{lmk}^{\chi}\right), \quad \text{(B.21)}$$

where we remember we are analyzing $l \geq 1$. From here we can find the canonical momenta, obeying canonical commutation relations, to be

$$\pi_{lmk}^{m} = \frac{\delta\mathcal{L}}{\delta\dot{A}_{lmk}^{m}} = \dot{A}_{lmk}^{m}{}^{*}, \qquad \pi_{lmk}^{\chi} = \frac{\delta\mathcal{L}}{\delta\dot{A}_{lmk}^{\chi}} = \left(1 - \frac{k^2}{\omega_{lk}^2 L^2}\right)\dot{A}_{lmk}^{\chi}{}^{*}, \qquad \text{(B.22)}$$

leading, by a Legendre transformation, to the Hamiltonian

$$\mathcal{H}_{lmk} = \frac{1}{2}\left(\pi_{lmk}^{m}{}^{*}\pi_{lmk}^{m} + \omega_{lk}^2 A_{lmk}^{m}{}^{*}A_{lmk}^{m} + \frac{\pi_{lmk}^{\chi}{}^{*}\pi_{lmk}^{\chi}}{\left(1 - \frac{k^2}{\omega_{lk}^2 L^2}\right)} + \left(\omega_{lk}^2 - \frac{k^2}{L^2}\right)A_{lmk}^{\chi}{}^{*}A_{lmk}^{\chi}\right). \qquad \text{(B.23)}$$

Note that in this notation $\mathcal{H}_{lmk} = \mathcal{H}_{l-m-k}$ and both contribute to the same mode. It will be useful to redefine the canonical variables as

$$\pi_{lmk}^{m} = \frac{1}{\sqrt{2}}\left(p_{lmk}^{(1)} - i\,p_{l-m-k}^{(1)}\right), \qquad \pi_{lmk}^{\chi} = \frac{1}{\sqrt{2}}\frac{\sqrt{l(l+1)}}{R}\left(q_{lmk}^{(2)} - i\,q_{l-m-k}^{(2)}\right), \qquad \text{(B.24)}$$

$$A_{lmk}^{m} = \frac{1}{\sqrt{2}}\left(q_{lmk}^{(1)} + i\,q_{l-m-k}^{(1)}\right), \qquad A_{lmk}^{\chi} = -\frac{1}{\sqrt{2}}\frac{R}{\sqrt{l(l+1)}}\left(p_{lmk}^{(2)} + i\,p_{l-m-k}^{(2)}\right), \qquad \text{(B.25)}$$

considering for the special case of $m = 0$ and $k = 0$ that

$$\pi_{l00}^{m} = p_{l00}^{(1)}, \qquad A_{l00}^{m} = q_{l00}^{(1)}, \qquad A_{lmk}^{\chi} = -\frac{R}{\sqrt{l(l+1)}}p_{l00}^{(2)}, \qquad \pi_{l00}^{\chi} = \frac{\sqrt{l(l+1)}}{R}q_{l00}^{(2)}. \quad \text{(B.26)}$$

This allows the rewriting of the Hamiltonian as the one corresponding to real harmonic oscillators with the same frequencies

$$\sum_{l=1}^{\infty}\sum_{m=-l}^{l}\sum_{k=-\infty}^{\infty}\mathcal{H}_{lmk} = \frac{1}{2}\sum_{l=1}^{\infty}\sum_{m=-l}^{l}\sum_{k=-\infty}^{\infty}\left((p_{lmk}^{(1)})^2 + (p_{lmk}^{(2)})^2 + \omega_{lk}^2(q_{lmk}^{(1)})^2 + \omega_{lk}^2(q_{lmk}^{(2)})^2\right). \quad \text{(B.27)}$$

The commutation relations, recovered from the canonical quantization of (B.21), simply take the form of $[q^{(1)}_{lmk}, p^{(1)}_{lmk}] = i$ and $[q^{(2)}_{lmk}, p^{(2)}_{lmk}] = i$.

Finally, we consider the case of $l = 0$. We further concentrate on the subset of those modes with $k \neq 0$, namely all the remaining modes but the one with $k = l = m = 0$. For such modes, we have the Lagrangian

$$\mathcal{L}_{00k} = \dot{A}^\chi_{00k}{}^* \dot{A}^\chi_{00k} + \frac{k^2}{L^2} A^0_{00k}{}^* A^0_{00k} - \frac{ik}{L}\left(\dot{A}^\chi_{00k}{}^* A^0_{00k} - A^0_{00k}{}^* \dot{A}^\chi_{00k}\right). \tag{B.28}$$

In this case, the Lagrange multiplier $A^0_{00k}$ produces the constraint $\dot{A}^\chi_{00k} = (ik/L)A^0_{00k}$, forcing $\mathcal{L}_{00k} = 0$.

Ultimately we are left with the mode $k = l = m = 0$, which turns out to be a free non-relativistic particle with Lagrangian and Hamiltonian given by the usual form

$$\mathcal{L}_{000} = \frac{1}{2}\dot{A}^\chi_{000}{}^2 \quad \Rightarrow \quad \mathcal{H}_{000} = \frac{1}{2}p_0^2, \qquad p_0 \equiv \dot{A}^\chi_{000}, \qquad q_0 \equiv A^\chi_{000}. \tag{B.29}$$

The canonical commutation relations for these modes are $[q_0, p_0] = i$. In summary, the full Hamiltonian takes the following form

$$\begin{aligned} H &= \frac{1}{2}\int dt \int_0^{2\pi} d\chi \int_0^\pi d\theta \int_0^{2\pi} d\varphi \sqrt{|g|}\, g^{ij}\left(E_i E_j + B_i B_j\right) \\ &= \frac{p_0^2}{2} + \frac{1}{2}\sum_{l=1}^\infty \sum_{m=-l}^l \sum_{k=-\infty}^\infty \left((p^{(1)}_{lmk})^2 + (p^{(2)}_{lmk})^2 + \omega^2_{lk}(q^{(1)}_{lmk})^2 + \omega^2_{lk}(q^{(2)}_{lmk})^2\right), \end{aligned} \tag{B.30}$$

where the gauge invariant fields write

$$\begin{aligned} E_i &= -p_0\left[\Phi^\chi_{000}\right]_i + \sum_{l=1}^\infty \sum_{m=-l}^l \sum_{k=0}^\infty [E_{lmk}]_i = -p_0\left[\Phi^\chi_{000}\right]_i + \sum_{l=1}^\infty\left[\omega_{l0}q^{(2)}_{l00}\left[\Phi^r_{l00}\right]_i - p^{(1)}_{l00}\left[\Phi^m_{l00}\right]_i\right] \\ &+ \sum_{l=1}^\infty \sum_{m=-l}^l \sum_{k=1}^\infty \left[\omega_{lk}q^{(2)}_{lmk}\left(\frac{\Phi^r_{lmk} + \Phi^{r*}_{lmk}}{\sqrt{2}}\right) + i\,\omega_{lk}q^2_{l-m-k}\left(\frac{\Phi^r_{lmk} - \Phi^{r*}_{lmk}}{\sqrt{2}}\right)\right] \\ &- \sum_{l=1}^\infty \sum_{m=-l}^l \sum_{k=1}^\infty \left[p^{(1)}_{lmk}\left(\frac{\Phi^m_{lmk} + \Phi^{m*}_{lmk}}{\sqrt{2}}\right) + i\,p^1_{l-m-k}\left(\frac{\Phi^m_{lmk} - \Phi^{m*}_{lmk}}{\sqrt{2}}\right)\right]. \end{aligned} \tag{B.31}$$

$$\begin{aligned} B_i &= \sum_{l=1}^\infty \sum_{m=-l}^l \sum_{k=0}^\infty [B_{lmk}]_i = \sum_{l=1}^\infty\left[\omega_{l0}q^{(1)}_{l00}\left[\Phi^r_{l00}\right]_i + p^{(2)}_{l00}\left[\Phi^m_{l00}\right]_i\right] \\ &+ \sum_{l=1}^\infty \sum_{m=-l}^l \sum_{k=1}^\infty \left[\omega_{lk}q^{(1)}_{lmk}\left(\frac{\Phi^r_{lmk} + \Phi^{r*}_{lmk}}{\sqrt{2}}\right) + i\,\omega_{lk}q^{(1)}_{l-m-k}\left(\frac{\Phi^r_{lmk} - \Phi^{r*}_{lmk}}{\sqrt{2}}\right)\right] \\ &+ \sum_{l=1}^\infty \sum_{m=-l}^l \sum_{k=1}^\infty \left[p^{(2)}_{lmk}\left(\frac{\Phi^m_{lmk} + \Phi^{m*}_{lmk}}{\sqrt{2}}\right) + i\,p^{(2)}_{l-m-k}\left(\frac{\Phi^m_{lmk} - \Phi^{m*}_{lmk}}{\sqrt{2}}\right)\right]. \end{aligned} \tag{B.32}$$

where we have conveniently changed the basis defining for $l \neq 0$

$$\left[\Phi^r_{lmk}(\theta,\varphi,\chi)\right]_i = \frac{1}{\omega_{lk}}E_{ijn}\nabla^j\left[\Phi^m_{lmk}(\theta,\varphi,\chi)\right]^n. \tag{B.33}$$

With these expressions, we can compute the electric and magnetic fluxes over the spatial sphere $S^2$. More concretely, the electric flux over $S^2$ surface $\Sigma$ defined by $\Sigma = \left\{ (t_0, \theta, \phi, \chi_0) \,\middle|\, \theta \in [0, \pi), \phi \in [0, 2\pi) \right\}$ is given by the integration of the pullback of $\star F$ to $\Sigma$

$$\Phi_E(\Sigma) = \int_\Sigma \star F = \int_0^\pi d\theta \int_0^{2\pi} d\phi \sqrt{|h|} E^\chi = -\left( \frac{L^{3/2}}{\sqrt{2}R} \right)^{-1} p_0(t_0). \tag{B.34}$$

We see that such electric flux is precisely the mode $p_0$, that commutes with the Hamiltonian and with all other operators except $q_0 \sim A^\chi_{000}$. This later variable, which is the canonical partner of the global electric flux on $S^2$, represents the global Wilson loop running parallel to the $S^1$ direction, that is, the result of integrating $A^\chi$ over the entire spatial manifold. The global magnetic flux over the $S^2$ surface $\Sigma$, however, is set to zero

$$\Phi_B(\Sigma) = \int_\Sigma F = \int_0^\pi d\theta \int_0^{2\pi} d\phi \sqrt{|h|} B^\chi = 0. \tag{B.35}$$

In this theory, there is a $U(1)$ electromagnetic duality symmetry acting on the modes different from the zero mode $l = 0, k = 0$, generated by the following charge

$$Q = \sum_{l=1}^\infty \sum_{m=-l}^l \sum_{k=0}^\infty \left( q^{(1)}_{lmk} p^{(2)}_{lmk} - q^{(2)}_{lmk} p^{(1)}_{lmk} \right). \tag{B.36}$$

This charge commutes with the Hamiltonian (B.30) and generates the following transformation on the fields

$$[Q, E^i_{lmk}] = i B^i_{lmk}, \quad [Q, B^i_{lmk}] = -i E^i_{lmk}. \tag{B.37}$$

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
