# Peer review of "ABJ anomaly as a U(1) symmetry and Noether's theorem"

_SciPost Physics, doi:SciPost Phys. 18, 041 (2025)_

## Round 1 · Referee Report · Anonymous (Referee 1) · 2024-6-5

Report

The manuscript discusses the symmetry participates in the ABJ anomaly such as chiral symmetry in QED4.

Here are several questions need to be clarified for the benefit of the readers:

  • In the ABJ anomaly, the current conservation is violated, and one can modify the current by a Chern-Simons term, which makes the current not gauge invariant. The author states that one can define gauge-invariant charge in (2.6), (2.39). However, the charge is not gauge-invariant, e.g. take the space to be S^2 x S^1 with flux on S^2 and perform gauge transformation by the angular variable of S^1. If the gauge group were R instead of U(1) the discussion would go through but not otherwise.

  • The authors state that there are no genuine non-invertible symmetries for dimension >2. Can the authors clarify how is the statement consistent with variety of counterexamples including free and interacting field theories and lattice models with well-behaved continuum limit? For instance, how are the symmetries generated by condensation defects in topological quantum field theories compatible with such a statement?

Recommendation

Ask for major revision

  • validity: -
  • significance: -
  • originality: -
  • clarity: -
  • formatting: -
  • grammar: -

Author:  Valentin Benedetti  on 2024-06-18  [id 4575]

(in reply to Report 1 on 2024-06-05)
Category:
answer to question

We thank the referee for the careful reading of the manuscript and for the valuable questions that can help clarify our presentation of the paper. Below we provide the answer to the referee's questions and suggest possible modifications that might help clarify the manuscript:

**Question 1:** The paper is focused on QFT in Minkowski space. In Minkowski space, the chiral charges constructed in (2.6) and (2.39) are gauge invariant and generate a U(1) symmetry. This invertibility of the group action is important to understand anomaly quantization in a new light, as a compatibility of the group action and the group of non-local operators. This is a new and important result of the paper. We propose to make this statement clearer in a new version of the paper by showing explicitly how a calculation of the anomaly results from Witten’s effect in section 2.3. Another important outcome of the present analysis is that the symmetry is a group, which makes it compatible with the Doplicher-Haag-Roberts (DHR) theorem. This theorem is a cornerstone in quantum field theory, that in our opinion, has been overlooked in recent literature. A consequence of this theorem is that internal symmetries in Minkowski space for d>2 are all determined by groups. Then, non-invertible symmetries cannot control the physics tested in accelerators such as the LHC.

However, the question posed by the referee is what happens for non-trivial topologies of the manifold. While this is not the focus of our paper, we devoted section 3 to such an analysis, particularly because the physics of the global modes in topologically non-trivial manifolds seems to be at the root of the terminology used in much of the recent literature. Our answer to this question is that, if the theory is defined by the gauge invariant physics in Minkowski space, there are in general several possible extensions of the theory to topologically non-trivial manifolds. Although these global choices can be understood as different possible manifestations of the symmetry in its global aspects, the local manifestations of the symmetry are the same for all choices and are always controlled by the same group. This is very different from what happens in d=2, where there are locally non-invertible symmetries, irrespective of global boundary conditions. Pointing out this fundamental difference in the meaning of non-invertibility of symmetry in QFT is the focus of our discussion in section 3. We propose to include a further paragraph explaining this difference better in a next version. As an example where all these ideas can be made completely explicit, we treated the free Maxwell field and its duality U(1) symmetry. The analysis shows there are many choices, including invertible ones, but the local symmetry is always U(1). Notice this symmetry is again considered non-invertible in the recent literature. Sections 3.2.3 and 3.4 serve as proof that this is not necessarily the case, and that any such non-invertible symmetry can be understood as a specific constraint/boundary condition on a theory with an invertible symmetry, and that moreover, such condition does not affect the local symmetry which is always U(1). This is not the case in d=2, a difference that is not appreciated in recent literature.

Though we analyzed several aspects of the discussion in the literature around non-invertible symmetries in d>2, we did not express our analysis in the language of large gauge transformations. This is what the referee asks in his first question. We propose to add such an explanation in a revised version. Essentially, invariance under a group of large gauge transformations may or may not be imposed according to the choice of the model in the topologically non-trivial manifold. These choices are equivalent to algebra choices for the global non local operators. 

The example of section 3.4 (Maxwell field on the S2xS1) was exactly described to make such an aspect transparent. For such model, the charge (3.22) generates a U(1) symmetry over local operators for any choice of boundary conditions and is invariant under the full group of all possible large gauge transformations. This charge controls the local physics and it clearly decouples from the charge generating the U(1) over the global mode (3.17) as in (3.25). However, the U(1) charge (3.25) acting on the global modes is not invariant under large gauge transformations. Nevertheless, this does not mean that such a charge does not generate a global symmetry. It all depends on how one defines the global aspects of the model. 

In a slightly different vein, to say that the gauge group for a free Maxwell field is U(1) or R does not have physical meaning except if the theory is considered in a topologically non-trivial manifold (or some other choice, such as artificially restricting the set of probes). In these manifolds, it is just a choice of global conditions/algebras. Every choice has the same local degrees of freedom given by the ordinary electric and magnetic fields.  To say that the gauge group is R or U(1) in this situation corresponds to what global non-local operators are chosen to exist or not in the model.  However, the existence of an electric charged field in QED is a physical local fact that makes the group of non-local operators (in compact topologically trivial regions) become a U(1) rather than R (as would be the case for the free Maxwell field). This is the focus of our paper, rather than what may be the case with global modes in a topologically non-trivial manifold. On the other hand, most of the literature has focused precisely on the global modes in order to determine the nomenclature and discuss the physics of generalized symmetries. We discuss this in section 3 following the advice of many colleagues. We think the standard approach based on global choices is motivated to a certain extent due to a lack of a clear understanding of the physical content of these generalized symmetries in the local physics. We supply this understanding by our realization that these generalized symmetries correspond to Haag duality violations for the local algebras of operators. 

**Question 2:** Notice our paper does not advocate in favor of the DHR theorem. This is a theorem. It holds under rather general conditions in continuum QFT. We just recall this theorem in the context of the present discussion. It is not our task to check models that claim the existence of a non-invertible symmetry in higher dimensions; we just put into focus that it would be their task, and an important and interesting one, to understand how this is compatible.  

However, we have certainly made an important effort to understand what is going on, and indeed clarify much of the tension between the different results. In particular, we have discussed explicitly in the paper what the claimed non-invertibility is about, and how it is compatible with the DHR theorem, for the ABJ anomaly or related cases. We also made everything completely explicit in a particular and simple example (duality for Maxwell field). 

In a like manner, we expect in most cases this compatibility will follow the same path as in the Maxwell model. We think people call non-invertible to something that is invertible regarding the local physics, say in Minkowski space, and the non-invertibility comes from global choices, for the global physics, in non-trivial topologies. Then, the choice of name “non-invertible” is just a choice of name, that may have its internal consistency depending on one’s interests and points of view. However, what we stress is that this terminology is quite misleading if two things are not simultaneously clarified: this non-invertibility is very different from what happens for non-invertible symmetries in d=2, and it is certainly not something that can be tested in accelerators, as usual internal symmetries. For the accelerator physics, all internal symmetries in d>2 have to be groups.  Some words can be added to the text of the paper to make it more transparent in this regard. 

On the other hand, it is also possible that some lattice models discussed in the literature may display local non-invertible symmetries in any dimension that do not make it to the continuum. This, of course, would also be something interesting to understand, but it lies outside of the scope of the present paper.

---

## Round 1 · Referee Report · Anonymous (Referee 2) · 2024-12-21

Strengths

The paper makes bold and unconventional claims that are rooted in an algebraic approach to QFT that dates back to work by Doplicher, Haag, and Roberts from 1969-1974, work that has been unknown to most of the QFT research community. The present paper work, and earlier works by the authors, claim that there is tension between a vast recent literature on generalized symmetries and claimed theorems from the algebraic approach. This tension could be interesting, and this work could this serve as a useful introduction to the algebraic approach, including claimed strong theorems. This paper considers the simple, concrete example of the ABJ anomaly and it makes numerous novel and surprising claims. Indeed, they claim that much recent literature on generalized symmetries is full of conceptual misunderstandings and in violation of algebraic QFT theorems. Such clashes could lead to a deeper understanding of QFT.

Weaknesses

The now-vast literature on generalized symmetries is based on an improved understanding of extended operators and global issues / distinctions. For example, it is now understood that there is a global difference between pure SU(2) gauge theory vs pure SO(3) gauge theory: the two theories have a different spectrum of Wilson (WL) and 't Hooft (TL) line operators. The present paper dismisses the improved understanding as "misconceptions" and wants to go back to the older perspective that ignores such observables and global issues / distinctions. For example, they say that the distinction between SU(2) vs SO(3) gauge theory is "misleading" and that they are the same theory with the same line operators; I believe that this is wrong. The authors state that they agree with the calculations in the literature, but disagree with the conceptual interpretations. That is my assessment of their paper: the calculations are correct but the interpretation suffers from their dismissing the recent literature as being full of misconceptions. I did not find their explanations of the claimed misconceptions to be sufficiently clear and compelling to be convinced that they are right about the literature being wrong. I agree with the points raised by the first referee and the author's response contains additional examples of their having what I believe are misconceptions: they claim "to say that the gauge group for a free Maxwell field is U(1) or R does not have physical meaning except if the theory is considered in a topologically non-trivial manifold (or..." The more modern understanding is that, even in flat spacetime, U(1) vs R Maxwell theory is quite different: the U(1) case admits monopole disorder operators and admits electric sources with either quantized electric charge or required sheets, whereas for R gauge group there are no monopole operators and no electric quantization condition. Nevertheless, I think the paper can be useful, provided that the reader views with caution their claims about the recent literature being full of misconceptions.

Report

The introduction is minimal, but clearly written. Section 2 considers "pion electrodynamics" aka axion-Maxwell theory. They state that they intentionally ignore the compactness of the pion/axion field even though, e.g. in footnote 5, they note that other literature tried similar things but dismissed them because of the compactness. The pion here can be regarded as the pseudo Nambu-Goldstone boson for the spontaneously broken axial U(1) (pseudo because of the ABJ anomaly), and the paper's dismissal of its compactness seems problematic. Relatedly, they attempt to define away the ABJ anomaly by redefining the current to include the non-gauge invariant term and they note that the associated charge is gauge invariant for gauge transformations or fields that vanish at infinity. Section 2.3 considers the non-spontaneously broken case of chiral symmetry in massless QED. The calculations appear to be correct, but again their point (which I think is incorrect) is that full axial U(1) symmetry is unbroken, because one can use the non-gauge invariant current that redefines away the ABJ anomaly. The claim is evidently that the ABJ anomaly should be reinterpreted as a good symmetry that has Hagg Duality Violation (HDV) classes of operators; HDV is not explained or summarized in the present paper, and one must read their earlier papers to understand the claim. Section 3 compares with other literature and this is the section that I see as having their own misconceptions, in their claims of misconceptions in the literature. However, I have to admit that I do not have a solid understanding of what they're saying with their formal equations in section 3.1 (the intention of trying to understand this, and the background in their earlier paper and the work of DHR was the cause of the excessive delay in this report). The other sections and appendices contain interesting comments, conjectures, and calculations. My overall assessment is that this paper contains interesting calculations and interesting claimed tensions with the algebraic results of DHR. I think that they are wrong about the literature being full of misconceptions, but I actually think that the paper should be published as-is, without further delay.

Requested changes

None

Recommendation

Publish (meets expectations and criteria for this Journal)

---

## Editorial Decision

published